# CONTROL-ORIENTED CLUSTERING OF VISUAL LATENT REPRESENTATION

**Han Qi**[*1]**, Haocheng Yin**[*†2] **and Heng Yang**[1]
[1]School of Engineering and Applied Sciences, Harvard University
[2]Department of Computer Science, ETH Zürich

## ABSTRACT

We initiate a study of the geometry of the visual representation space —the information channel from the vision encoder to the action decoder— in an image-based control pipeline learned from behavior cloning. Inspired by the phenomenon of *neural collapse* (NC) in image classification (Papyan et al., 2020), we empirically demonstrate the prevalent emergence of a similar *law of clustering* in the visual representation space. Specifically,

- In *discrete* image-based control (e.g., Lunar Lander), the visual representations cluster according to the natural discrete action labels;
- In *continuous* image-based control (e.g., Planar Pushing and Block Stacking), the clustering emerges according to "*control-oriented*" classes that are based on (a) the relative pose between the object and the target in the input or (b) the relative pose of the object induced by expert actions in the output. Each of the classes corresponds to one *relative pose orthant* (REPO).

Beyond empirical observation, we show such a law of clustering can be leveraged as an *algorithmic tool* to improve test-time performance when training a policy with limited expert demonstrations. Particularly, we *pretrain* the vision encoder using NC as a *regularization* to encourage control-oriented clustering of the visual features. Surprisingly, such an NC-pretrained vision encoder, when finetuned end-to-end with the action decoder, boosts the test-time performance by $10\%$ to $35\%$. Real-world vision-based planar pushing experiments confirmed the surprising advantage of control-oriented visual representation pretraining. [1]

## 1 INTRODUCTION

We use a toy example to (a) introduce the concept of neural collapse and the task of policy learning from expert demonstrations, and (b) synchronize readers from the respective communities.

**Minimum-time double integrator** Consider a dynamical system known as the *double integrator*

$$\ddot{q}(t) = u(t), \tag{1}$$

where $q \in \mathbb{R}$ is the position, and $u \in \mathbb{U} := [-1, 1]$ is the external control that decides the system's acceleration. For an example, imagine $q$ as the position of a car and $u$ as how much throttle or braking is applied to the car. Let $x(t) := (q(t), \dot{q}(t)) \in \mathbb{R}^2$ be the state of the system. Suppose the system starts at $x(0) = \chi$, and we want to find the optimal state-feedback policy that drives the system to the origin using *minimum time*. Formally, this is an optimal control problem written as

$$\min_{u(t)} T, \quad \text{subject to} \quad x(0) = \chi, \ x(T) = 0, \ u(t) \in \mathbb{U}, \forall t, \ \text{and (1)}. \tag{2}$$

Problem (2) admits a closed-form optimal policy (Rao & Bernstein, 2001):

$$u_\star = \pi_\star(x) := \begin{cases} +1 & \text{if } \left(\dot{q} < 0 \text{ and } q \le \frac{1}{2}\dot{q}^2\right) \text{ or } \left(\dot{q} \ge 0 \text{ and } q < -\frac{1}{2}\dot{q}^2\right) \\ 0 & \text{if } q = 0 \text{ and } \dot{q} = 0 \\ -1 & \text{otherwise.} \end{cases} \tag{3}$$

This optimal policy is *bang-bang*: it applies either full throttle or full brake until reaching the origin.

---

[*]Equal contribution
[†]Work done during visit at the Harvard Computational Robotics Lab
[1]https://computationalrobotics.seas.harvard.edu/ControlOriented_NC

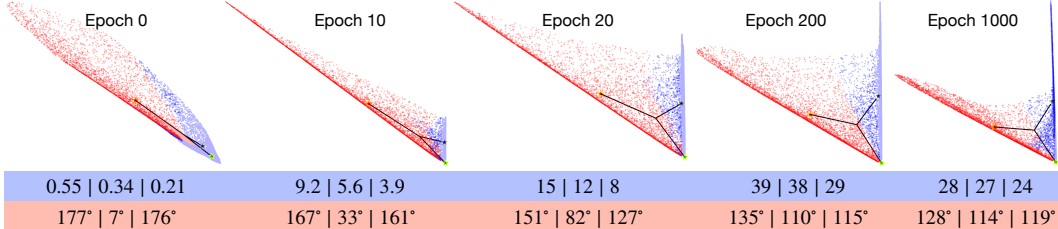

| Epoch 0 | Epoch 10 | Epoch 20 | Epoch 200 | Epoch 1000 |
|---|---|---|---|---|
| 0.55 \| 0.34 \| 0.21 | 9.2 \| 5.6 \| 3.9 | 15 \| 12 \| 8 | 39 \| 38 \| 29 | 28 \| 27 \| 24 |
| 177° \| 7° \| 176° | 167° \| 33° \| 161° | 151° \| 82° \| 127° | 135° \| 110° \| 115° | 128° \| 114° \| 119° |

Figure 1: Per-class (red, blue, green) globally centered features (points) and mean vectors (black lines with $\star$ endpoints) from the penultimate latent space when cloning the optimal bang-bang policy (3) from expert demonstrations. Numbers in the blue band represent the lengths of the three per-class mean vectors, and numbers in the red band represent the angles spanned by pairs of per-class mean vectors. The lengths and angles tend to be equal to each other as training progresses.

**Behavior cloning**  We now try to learn the optimal policy from data, using a strategy called *behavior cloning* (Torabi et al., 2018). We collect $3N$ samples (a.k.a. expert demonstrations) from the optimal policy $\pi_\star(x)$; each batch of $N$ samples have "label" $+1$, $0$, and $-1$, respectively. Then we treat policy learning as a classification problem given input $x$ and output $u$. We design a six-layer MLP ($2 \rightarrow 64 \rightarrow 64 \rightarrow 64 \rightarrow 3 \rightarrow 3$) and train it with the cross-entropy loss. We use $N = 5000$. For the $u = 0$ class, we repeat $N$ times the sample "$x = 0$" to balance the dataset.

**Geometry of the representation space**  The MLP is able to learn a good policy, but this is not the purpose of our experiment. We instead look at the geometry of the three-dimensional feature space at the penultimate layer. Particularly, let $\{f_i^{+1}\}_{i=1}^N$, $\{f_i^0\}_{i=1}^N$, $\{f_i^{-1}\}_{i=1}^N$ be the three sets of feature vectors corresponding to controls $\{+1, 0, -1\}$. We compute the per-class and global mean vectors:

$$\mu^c = \frac{1}{N} \sum_{i=1}^N f_i^c, \ \ c = +1, 0, -1; \quad \mu = \frac{1}{3} \left( \mu^{+1} + \mu^0 + \mu^{-1} \right). \tag{4}$$

Let $\tilde{\mu}^c := \mu^c - \mu$ be the *globally-centered* class mean for each class $c$. Fig. 1 plots the class means $\tilde{\mu}^c$ and the globally-centered feature vectors $\tilde{f}_i^c := f_i^c - \mu$ in different colors, as training progresses. We observe a clear *clustering* of the features according to their labels. Moreover, the clustering admits a precise geometry: (a) the lengths of globally-centered class means $\|\tilde{\mu}^c\|$ tend to be equal to each other for $c = +1, 0, -1$, and (b) the angles spanned by pairs of class mean vectors $\angle(\tilde{\mu}^{c_1}, \tilde{\mu}^{c_2})$ also tend to be equal to each other for $c_1 \neq c_2$, shown by the perfect "tripod" in Fig. 1 epoch 1000.

**Neural collapse**  The clustering phenomenon shown in Fig. 1 was first observed by Papyan et al. (2020) in image classification and dubbed the name *neural collapse* (NC). In particular, NC refers to a set of four manifestations in the representation space (i.e., the penultimate layer):

(NC1)  Variability collapse: feature vectors of the same class converge to their class mean.

(NC2)  Simplex ETF: globally-centered class mean vectors converge to a geometric configuration known as Simplex Equiangular Tight Frame (ETF), i.e., mean vectors have the same lengths and form equal angles pairwise (as shown in Fig. 1).

(NC3)  Self-duality: the class means and the last-layer linear classifiers are self-dual.

(NC4)  Nearest class-center prediction: the network predicts the class whose mean vector has the minimum Euclidean distance to the feature of the test image.

Since its original discovery, NC has attracted significant interests, both empirical (Jiang et al., 2023; Wu & Papyan, 2024; Rangamani et al., 2023) and theoretical (Fang et al., 2021; Han et al., 2021); see Appendix I for a detailed discussion of existing literature. The results we show in Fig. 1 are just another example to reinforce the prevalence of neural collapse, because behavior cloning of the bang-bang optimal controller reduces to a classification problem.

**Our goal**  Does a similar *law of clustering*, in the spirit of NC, happen when cloning image-based control policies? Two motivations underlie this question. First, understanding the structure of learned representation has been a fundamental pursuit towards improving the interpretability of deep

learning models. While the discovery of NC has deepened our understanding of visual representation learning for *classification*, such an understanding is missing when using visual representation for *decision-making*. Second, just as related work has shown the benefits of NC for generalization and robustness (Bonifazi et al., 2024; Liu et al., 2023; Yang et al., 2022), we seek to uncover new algorithmic tools that improve model performance by "shaping" the latent representation space.

Towards this goal, we consider a general image-to-action architecture consisting of (a) a *vision encoder* that embeds high-dimensional images as compact visual features (He et al., 2016; Oquab et al., 2023), and (b) an *action decoder* that generates control outputs given the latent vectors (Mandlekar et al., 2021; Chi et al., 2023). The entire pipeline is trained end-to-end using expert demonstrations. We study the instantiation of this architecture in three tasks: Lunar Lander from OpenAI Gym (Brockman, 2016), Planar Pushing that is popular in robotics (Chi et al., 2023), and Block Stacking from MimicGen (Mandlekar et al., 2023). The first is a *discrete control* task, while the second and the third are *continuous control* tasks. Fig. 2 overviews the architecture and the tasks.

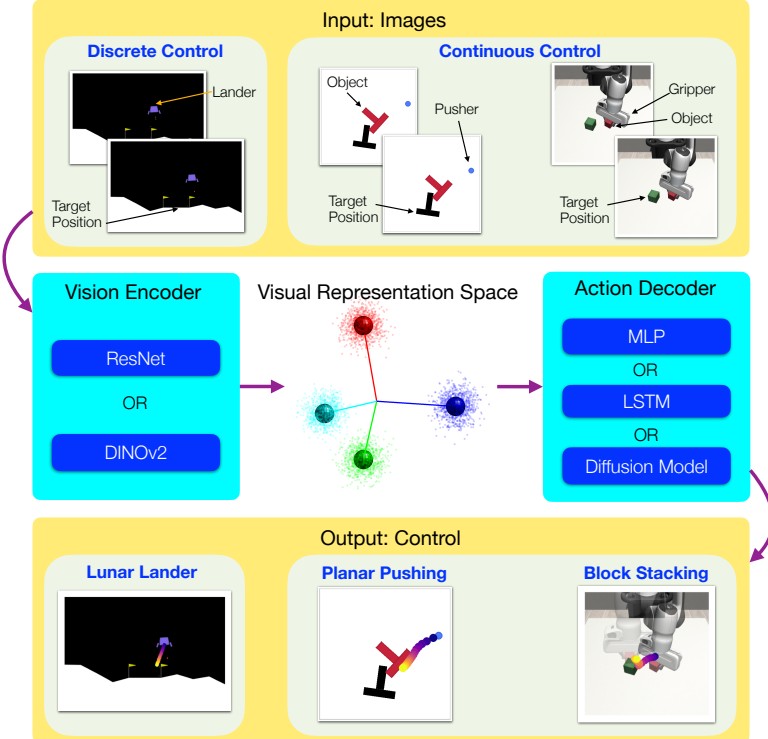

Figure 2: Investigation of a law of clustering, similar to NC, in the visual representation space.

The bridge and information channel from vision to control is the visual representation space. We aim to study the geometry of the visual representation space from the lens of neural collapse. Particularly, we seek to answer two fundamental questions.

(Q1) Does a clustering phenomenon similar to NC happen in the visual representation space? If so, according to which "classes" do they cluster?

(Q2) Is the extent to which neural collapse happens related to the model's test-time performance? If so, can we transform neural collapse from a "phenomenon" to an "algorithmic tool"?

These two questions have never been answered before —either empirically or theoretically— because vision-based control is in stark contrast with existing literature in neural collapse for image classification. First, a prerequisite of NC is that the training data need to already be classified. Continuous vision-based control tasks (like planar pushing and block stacking), however, are *regression* problems where the output is a continuous control signal. There is no supervision coming from classification whatsoever. Second, theoretical analysis of NC typically assumes a linear classifier from the representation space to the model output (Han et al., 2021; Fang et al., 2021; Jiang et al.,

2023). This assumption is strongly violated in vision-based control because from visual representation space to control lies a rather nonlinear and complicated action decoder (see Fig. 2).

**Our contribution**  Despite the challenges mentioned, we empirically demonstrate that the answers to both questions are affirmative in vision-based control tasks. Our contributions are:

(C1) **Control-oriented clustering**  We first show that, on the discrete-control task Lunar Lander, visual latent features cluster according to the discrete action labels. This forms the natural vision-based generalization of the double integrator example shown in Fig. 1. We then primarily focus on continuous-control tasks. A natural path to study NC for a regression problem is to give each sample a class label and check whether the visual features cluster according to the labels. Then, the nontrivial question becomes what should these classes be? Based on the posit that the visual representation should *convey a goal of control* for the action decoder, we design two "control-oriented" classification strategies. They compute (a) the relative pose between the object and the target in the input image space, or (b) the relative pose change of the object induced by the sequence of expert actions, and classify the samples into 8 classes, each corresponding to one orthant in the relative pose space (called a REPO). We then demonstrate the prevalent emergence of neural collapse in the visual representation space, in both Planar Pushing and Block Stacking.

(C2) **Control-oriented visual representation pretraining**  When the number of expert demonstrations is decreased, the strength of NC decreases, so does the model's test-time performance. This motivates us to leverage NC as an algorithmic tool to improve the model's performance under insufficient demonstrations. Indeed, we show that by using the control-oriented NC metrics as the loss function to pretrain the vision encoder, we obtain $10\%$ to $35\%$ boost in the model's test-time performance. Real-world Planar Pushing experiments confirmed the advantage of NC pretraining.

**Paper organization**  We first show observation of neural collapse in the discrete-control task Lunar Lander in §2. We then introduce the problem setup of continuous vision-based control tasks, control-oriented classification, and the prevalence of neural collapse in the visual representation space for continuous-control tasks in §3. We describe our method of NC pretraining and show it improves model performance in §4. We demonstrate real-world robotic experiments in §5 and conclude in §6.

## 2  NEURAL COLLAPSE IN DISCRETE VISION-BASED CONTROL

As a transition from our *bang-bang* policy in §1 to more challenging continuous vision-based control tasks, we first study whether the visual representation clusters according to the given discrete action labels in a discrete vision-based control task, *Lunar Lander* (Brockman, 2016). In a nutshell, Lunar Lander is a task where the lander aims to reach a given target position by deciding between four discrete actions (see Fig. 2). We first train a performant *state-based* policy using reinforcement learning (RL) and then use the RL expert to collect vision-based demonstrations for behavior cloning (BC). The BC pipeline uses ResNet18 as the vision encoder and an MLP as the action decoder, supervised by cross-entropy loss with the expert demonstrations. Appendix A provides more details.

Since the action space for Lunar Lander is discrete with 4 actions, we directly study the clustering phenomenon of the latent features according to these 4 actions. Suppose the training dataset contains $M$ samples of images (input) and actions (output), and denote the set of visual features as $\mathcal{F} = \{f_t\}_{t=1}^{M}$, where each feature $f_t$ is computed by passing the input images through the ResNet18 encoder. We assign the action label $c \in [4]$ to each feature $f_t$ and denote it $f_t^c$.

**Neural collapse metrics**  We compute three metrics to evaluate (NC1) and (NC2). Define

$$\mu^c = \frac{1}{M_c} \sum_{i=1}^{M_c} f_i^c, c = 1, \ldots, C, \quad \mu = \frac{1}{M} \sum_{c=1}^{C} \sum_{i=1}^{M_c} f_i^c, \tag{5}$$

as the class mean vectors and the global mean vector, respectively. Note that $M_c$ denotes the total number of samples in each class $c$ and $\sum_{c=1}^{C} M_c = M$. Then define $\tilde{\mu}^c := \mu^c - \mu$ as the globally-centered class means, and $\tilde{f}_i^c := f_i^c - \mu$ as the globally-centered feature vectors. Consistent with Wu

& Papyan (2024), we evaluate (NC1) using the *class-distance normalized variance* (CDNV) metric that depends on the ratio of within-class to between-class variabilities:

$$\text{CDNV}_{c,c'} := \frac{\sigma_c^2 + \sigma_{c'}^2}{2\|\tilde{\mu}^c - \tilde{\mu}^{c'}\|^2}, \quad \forall c \neq c', \tag{6}$$

where $\sigma_c^2 := \frac{1}{M_c-1} \sum_{i=1}^{M_c} \|\tilde{f}_i^c - \tilde{\mu}^c\|^2$ is the within-class variation. Clearly, (NC1) happens when $\text{CDNV}_{c,c'} \to 0$ for any $c \neq c'$. We use a single number CDNV to denote the mean of all $\text{CDNV}_{c,c'}$ for $c \neq c'$. We evaluate (NC2) using the standard deviation (STD) of the lengths and angles spanned by $\tilde{\mu}^c$ (calling AVE as the shortcut for averaging):

$$\text{STDNorm} := \frac{\text{STD}\left(\{\|\tilde{\mu}^c\|\}_{c=1}^C\right)}{\text{AVE}\left(\{\|\tilde{\mu}^c\|\}_{c=1}^C\right)}, \quad \text{STDAngle} := \text{STD}\left(\left\{\left\langle \frac{\tilde{\mu}^c}{\|\tilde{\mu}^c\|}, \frac{\tilde{\mu}^{c'}}{\|\tilde{\mu}^{c'}\|}\right\rangle\right\}_{c \neq c'}\right). \tag{7}$$

Clearly, (NC2) happens if and only if both STDNorm and STDAngle become zero. We do not evaluate (NC3) and (NC4) because they require a linear classifier from the representation space to the output, which does not hold in our vision-based control setup.

**Results**  Fig. 3 plots the three NC evaluation metrics w.r.t. training epochs for Lunar Lander. We observe consistent decrease of three NC metrics as training progresses and approaching zero at the end of training, suggesting strong control-oriented clustering in the visual representation space.

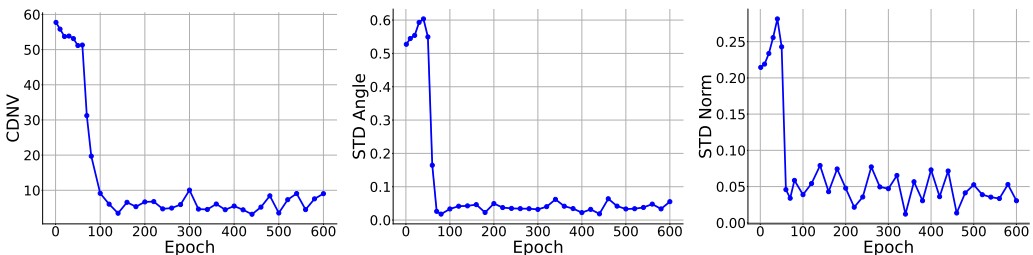

Figure 3: Emergence of neural collapse in the visual representation space for Lunar Lander. From left to right, it shows three NC metrics w.r.t. training epochs using ResNet18 as the vision encoder and an MLP as the action decoder, with four discrete actions as classification labels.

## 3 NEURAL COLLAPSE IN CONTINUOUS VISION-BASED CONTROL

We now consider continuous vision-based control tasks, where an obvious set of class labels does not exist anymore. We focus on two tasks: Planar Pushing and Block Stacking.

Planar pushing (Lynch & Mason, 1996; Yu et al., 2016) is a longstanding problem in robotics, and consists of controlling a *pusher* to push a given *object* to follow certain trajectories or to a target position. This problem is fundamental because manipulating objects is crucial to deploy robots in the physical world (Bicchi & Kumar, 2000). Despite how effortlessly humans perform this task, planar pushing represents one of the most challenging problems for classical model-based control (e.g., optimization-based control (Wensing et al., 2023)) due to the underlying hybrid dynamics and underactuation (i.e., the controller needs to plan where/when the pusher should make/break contact with the object and whether to slide along or stick to the surface of the object, leading to different "modes" in the dynamics (Goyal et al., 1991; Hogan & Rodriguez, 2020)).

Block stacking (Mandlekar et al., 2023; Zhu et al., 2020) is a classical manipulation task, aiming to stack one block onto another target block. This involves lifting up a block, moving the block above the target, rotating the block to match the angle of the target, and putting the block down.

For both tasks, we describe how to train a vision-based control policy from expert demonstrations.

**Policy learning from demonstrations**  We are given a collection of $N$ expert demonstrations $\mathcal{D} = \{D_i\}_{i=1}^N$ where each $D_i$ is a sequence of images and controls

$$D_i = (I_0, u_0, I_1, u_1, \ldots, I_{l_i-1}, u_{l_i-1}, I_{l_i}),$$

with $I$ the image, $u$ the control, $l_i$ the length of $D_i$, and in the final image $I_{l_i}$ the object is moved to the target position. From $\mathcal{D}$ we extract a set of $M$ training samples $\mathcal{S} = \{s_t\}_{t=1}^M$ where each sample $s_t$ consists of a sequence of $K$ images and a sequence of $H$ controls

$$s_t = (I_{t-K+1}, \ldots, I_{t-1}, I_t \mid u_t, u_{t+1}, \ldots, u_{t+H-1}). \tag{8}$$

$M$ is usually much larger than $N$. The end-to-end policy is then trained on $\mathcal{S}$, which takes as input the sequence of images and outputs the sequence of controls. At test time, the policy is executed in a receding horizon fashion (Mayne & Michalska, 1988) where only the first predicted control $u_t$ is executed and then a new sequence of controls is re-predicted to incorporate feedback from the new image observations. For every training sample (8), denote

$$f_t = \text{VisionEncoder}(I_{t-K+1}, \ldots, I_{t-1}, I_t) \tag{9}$$

as the visual feature of the sequence of images.

## 3.1 Classification according to Relative Pose Orthants (Repos)

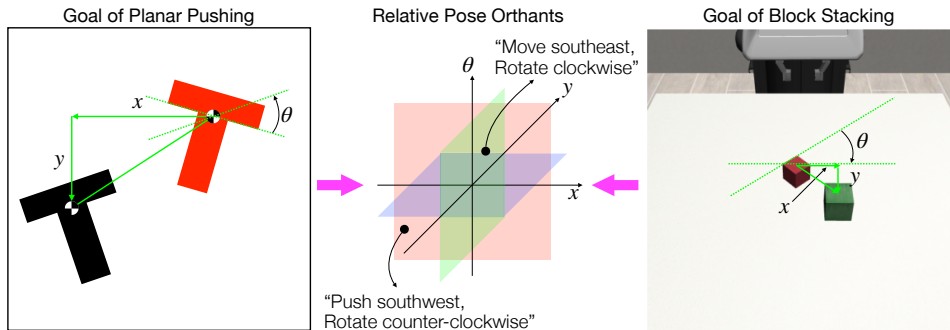

Figure 4: Control-oriented classification for continuous vision-based control tasks. Left: planar pushing; Right: block stacking.

The essence of neural collapse is a *law of clustering* in the representation space. To study this, we need to assign every feature vector $f_t$ a class. In image classification and discrete vision-based control (e.g., Lunar Lander), this class is explicitly given. However, in vision-based continuous control the output is a sequence of controls, what should the "class" be?

A natural choice is to perform $k$-means clustering of the output actions. Unfortunately, not only is this classification not interpretable, it also does not lead to observation of NC (see Appendix B).

We then conjecture that for control tasks such as planar pushing and block stacking, the role of the vision encoder is to convey a "control goal" to the action decoder from image observations. For example, looking at the left image in Fig. 4 for Planar Pushing, the vision encoder may set the goal of control to "push the T block southwest and rotate it counter-clockwise". Similarly, in the right image in Fig. 4 for Block Stacking, the vision encoder may convey "move the block southeast and rotate it clockwise". Building upon this intuition, we design two strategies to classify each training sample: one that is *goal-based* and the other that is *action-based*. Since both strategies lead to similar observations of NC, we only present goal-based classification in the main text and refer the interested reader to Appendix C for action-based classification.

**Goal-based classification** Given a sample $s_t$ as in (8), we look at image $I_t$. We compute the relative pose of the target position with respect to the object. As depicted in Fig. 4, this relative pose is a triplet $(x, y, \theta)$ containing a 2D translation and a 1D rotation for both planar pushing and block stacking. We divide the 3D space that the relative pose triplet $(x, y, \theta)$ lives in into eight classes based on the signs of $x, y, \theta$. In other words, each class corresponds to one orthant of the space (called a *relative pose orthant*, or in short a REPO), as visualized in Fig. 4 middle. A nice property is that the resulting classes are semantically interpretable!

**Remark 1** (Finegrained REPOs). *What will happen if the relative pose space is divided into a larger number of classes? In Appendix G.4, we divide the relative pose space into 64 and 216 classes, and demonstrate that such a law of clustering still holds, albeit to a slightly weaker extent.*

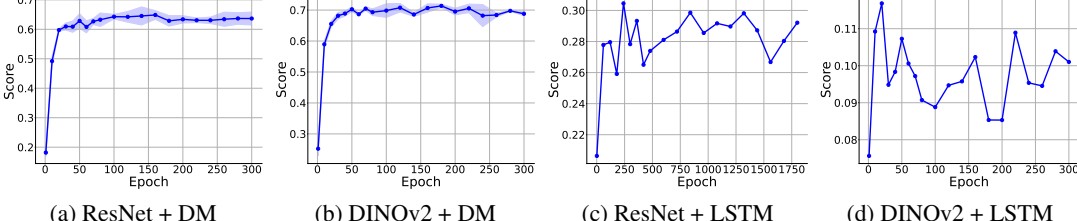

| (a) ResNet + DM | (b) DINOv2 + DM | (c) ResNet + LSTM | (d) DINOv2 + LSTM |

Figure 5: Test scores w.r.t. training epoches of four different instantiations of the image-based control pipeline for planar pushing. In (a) and (b) we show test scores of three random seeds. In (c) and (d) we show test scores of a single seed because using LSTM as the action decoder leads to poor test-time performance, an observation that is consistent with Chi et al. (2023).

## 3.2 PREVALENT EMERGENCE OF NEURAL COLLAPSE

Using the control-oriented classification strategy described above, we are ready to study whether a law of clustering similar to NC emerges in the visual representation space for continuous control tasks. Given the set of visual features $\mathcal{F} = \{f_t\}_{t=1}^{M}$, we assign a class label $c \in [C]$ to each feature vector $f_t$ and denote it $f_t^c$. We used the same neural collapse metrics introduced in §2.

### 3.2.1 PLANAR PUSHING

**Simulation setup**  We collect $N = 500$ expert demonstration on a push-T setup shown in Fig. 2. At each round, the object and target positions are randomly initialized and the same human expert controls the pusher to push the object into alignment with the target position (through a computer interface provided by `pymunk` (Blomqvist, 2024)). This provides $M = 55,480$ training samples. We train four different instantiations of the image-based control pipeline: using ResNet or DINOv2 as the vision encoder, and Diffusion Model (DM) or LSTM as the action decoder. The four trained models are evaluated on a test push-T dataset with 100 tasks. We define our evaluation metric as the ratio of the overlapping area between the object and the target to the total area of the target.

**Results**  Fig. 5 shows the evaluation scores of four different models. DINOv2 combined with a diffusion model (DM) attains the best performance around $70\%$. ResNet combined with DM (the original diffusion policy from Chi et al. (2023)) is slightly worse but quite close. When an LSTM is used as the action decoder to replace DM, the performance significantly drops, an observation that is consistent with Chi et al. (2023) and confirms the advantage of using DM as the action encoder. For this reason, it is not worthwhile training models with LSTM from different random seeds.

Fig. 6 plots the three NC evaluation metrics w.r.t. training epochs for two different models (both with DM) with goal-based classification strategy described in §3.1. We observe consistent decrease of three NC metrics as training progresses, suggesting the prevalent emergence of a law of clustering that is similar to NC. Results with action-based classification are similar and provided in Appendix G.2. Despite the poor test-time performance of trained models using LSTM, similar neural collapse is observed and shown in Appendix G.

**From pretrained to finetuned ResNet features**  As observed in Chi et al. (2023); Kim et al. (2024); Team et al. (2024), *pretrained* ResNet features deliver poor performance when used for control (such as planar pushing) and end-to-end finetuning is necessary to adapt the vision features (see Appendix D for concrete evidence). This is intriguing: why are pretrained visual features insufficient for control? We now have a plausible answer. ResNet is pretrained for image classification, and according to NC, the pretrained ResNet features are clustered according to the class labels in image classification, such as dogs and cats. However, under our NC observation in Fig. 6, ResNet features for planar pushing are clustered according to "control-oriented" classes that are related to the relative pose between the object and the target. Therefore, during finetuning, we conjecture the visual features have "re-clustered" according to the new task. We verify this conjecture in Appendix E.

### 3.2.2 BLOCK STACKING

**Simulation setup**  Designed by Mandlekar et al. (2023), the block stacking is implemented as one of the manipulation tasks in MimicGen dataset using robosuite framework (Zhu et al., 2020)

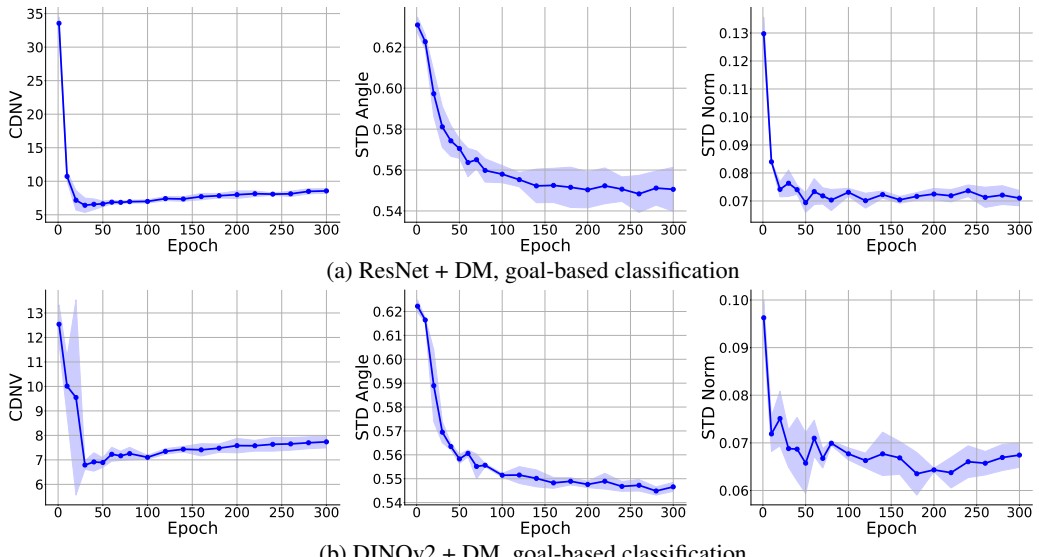

(a) ResNet + DM, goal-based classification

(b) DINOv2 + DM, goal-based classification

Figure 6: Prevalent emergence of neural collapse in the visual representation space for planar pushing. (a) Three NC metrics w.r.t. training epochs using ResNet as the vision encoder and diffusion model (DM) as the action decoder, with goal-based classification label. (b) Three NC metrics w.r.t. training epochs using DINOv2 as the vision encoder and DM as the action decoder. All the plots show the mean and standard deviation (in shaded band) over three random seeds. The test scores are shown in Fig. 5(a)(b). Similar observations of NC hold when replacing DM with LSTM as the action decoder, shown in Appendix G.

backended on MuJoCo (Todorov et al., 2012). To train the behavior cloning pipeline, we used the dataset "core stack_d0" provided by MimicGen which contains $N = 1000$ demos as our expert demonstrations. This provides $M = 107,590$ training samples. We used ResNet18 as the vision encoder and diffusion model as the action decoder.

**Results** Fig. 7 plots the three NC evaluation metrics w.r.t. training epochs for Resnet+DM with the goal-based classification strategies described in §3.1. We observe consistent decrease of three NC metrics as training progresses, suggesting the prevalent emergence of a law of clustering that is similar to NC in the block stacking manipulation task.

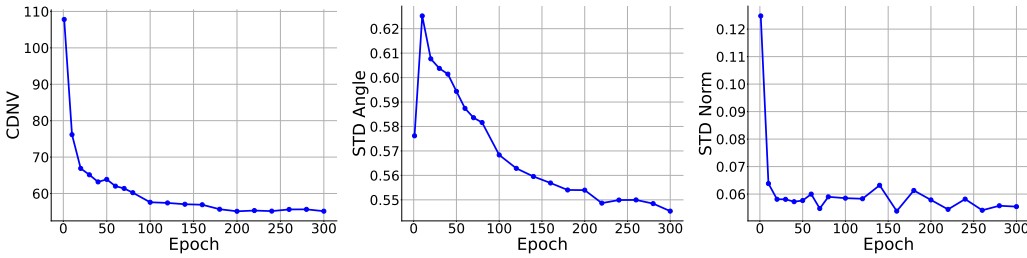

Figure 7: Prevalent emergence of neural collapse in the visual representation space for Block Stacking. From left to right, it shows three NC metrics w.r.t. training epochs using ResNet as the vision encoder and a DM as the action decoder, with goal-based classification label.

# 4 VISUAL REPRESENTATION PRETRAINING WITH NEURAL COLLAPSE

Our experiments in §3.2 empirically demonstrated that an image-based control policy trained on sufficient amount of data exhibits an elegant control-oriented law of clustering in its visual representation space. We now study whether such clustering still holds under insufficient data.

| Pushing tasks | 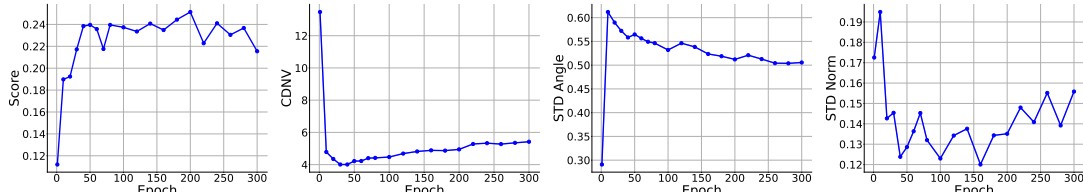 | | | |
|---|---|---|---|---|
| Baseline | 0.202 | 0.423 | 0.385 | 0.241 |
| NC-pretrained (goal-based) | 0.360 | 0.720 | 0.504 | 0.347 |
| NC-pretrained (action-based) | 0.378 | 0.769 | 0.488 | 0.351 |

Table 1: Pretraining a ResNet vision encoder by minimizing the NC metrics significantly improves test-time model performance on four domains corresponding to four letters in the word "ROBOT".

**Weak clustering**  We use the same setup as in §3.2.1, focus on the (ResNet+DM) model, and train it using only 100 demonstrations. Fig. 8 shows the test-time performance and three NC metrics according to the goal-based classification. Neural collapse appears to still hold, but to a much weaker extent, particularly when looking at the STDNorm metric.

Figure 8: Weak neural collapse under 100 demonstrations of planar pushing. NC metrics are calculated using goal-based classification. See Appendix G for results using action-based classification.

**Visual pretraining**  If we knew a well-trained model exhibits strong clustering in its visual representation space, how about we encourage such a phenomenon even without sufficient data? To test this idea, we pretrain the vision encoder by explicitly minimizing the three NC metrics ((6) and (7)) prior to training the vision encoder jointly with the action decoder. Appendix F presents more details. Table 1 presents the comparison of test-time performance between the baseline model and models whose vision encoder is pretrained to encourage control-oriented clustering. We observe a substantial improvement by using NC pretraining: the minimum improvement is around $10\%$ on the letter B, and the maximum improvement is around $35\%$ on the letter O (we intentionally carved out a piece to break symmetry). Note that all models are trained using only 100 expert demonstrations!

## 5  REAL-WORLD VALIDATION

The improvement shown in Table 1 is astonishing and motivates us to further validate the effectiveness of NC pretraining in the real world.

**Setup**  Fig. 9 depicts our real-world setup for vision-based planar pushing. The pusher is attached to the end-effector of a Franka Panda robotic arm to push a T block into the target position on a flat surface. An overhead Intel RealSense camera is used to provide image observations to the pusher. To enable NC pretraining in the real world, we need groundtruth relative pose of the object w.r.t. the target position. To provide reliable estimate of the relative pose, we attached an AprilTag (Olson, 2011) to the bottom of the T-block, which enables real-time accurate pose estimation. To collect real-world expert demonstrations, we follow Chi et al. (2023) and use a SpaceMouse to teleoperate the robotic arm to push the T block. We collected 100 demonstrations where the initial position of the T-block is randomized at each run, and this took around 10 hours. We trained two policies: (a) a baseline that is the (ResNet+DM) model trained directly using 100

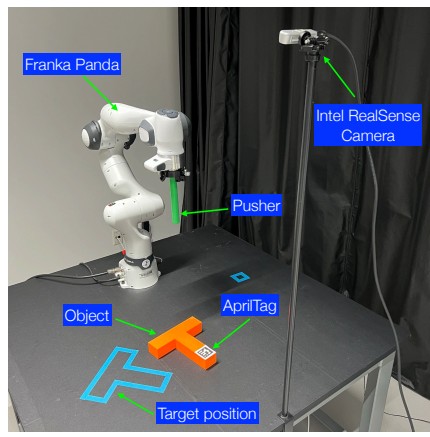

Figure 9: Real-world setup.

demonstrations, and (b) an NC-pretrained model whose ResNet is first pretrained by minimizing NC metrics and then jointly trained with DM.

**Visual complexities**  Real-world robotic manipulation needs to handle visually complex scenes. Therefore, we created another visually challenging setup where 11 distracting objects and 2 well-known paintings are placed in the workspace (see Fig. 10(b)). For this challenging setup, we collected 200 expert demonstrations because 100 demonstrations were insufficient to train robust policies. Using this dataset, we trained two policies with the same network setup as described before.

**Results**  We tested both models on a set of 10 new push-T tasks where the T block is initialized at positions not seen during training. We intentionally tried our best to initialize the T block at exactly the same position for fair comparison of two models (the reader can check this in Fig. 10). For the clean background setup, the baseline model succeeded 5 out of 10 times, and the NC-pretrained model succeeded 8 out of 10. For the visually complex background setup, the baseline model succeeded 6 out of 10, and the NC-pretrained model succeeded 9 out of 10. Fig. 10 shows two tests comparing the trajectories of the baseline and the NC-pretrained models under different background setups. The rest of the tests are shown in Appendix H and supplementary videos.

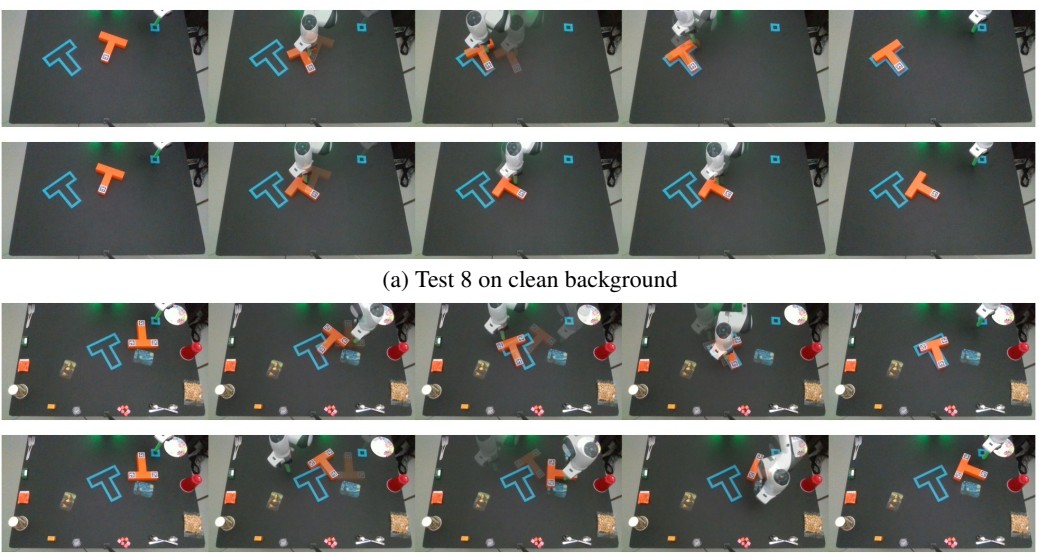

(a) Test 8 on clean background

(b) Test 3 on visually complex background

Figure 10: Two real-world tests. In every test, top row shows trajectories of NC-pretrained model and bottom row shows trajectories of baseline. The first and last column show the initial and final position of the object, respectively. The middle three columns show overlaid trajectories generated by the policies. Full results of 10 tests are shown in Appendix H and supplementary videos.

## 6  CONCLUSION

We presented an empirical investigation of the geometry of the visual representation space in an end-to-end image-based control policy for both discrete and continuous environments. We demonstrated that, similar to the phenomenon of neural collapse in image classification, a prevalent control-oriented law of clustering emerges in the visual representation space. Further, this law of clustering can be effectively leveraged to pretrain the vision encoder for improved test-time performance.

**Limitations and future work**  Our research opens lots of future research directions, such as theoretical analysis of control-oriented clustering, extension of such investigation to policies trained from reinforcement learning, and connections to neuroscience. A major limitation of our work is the observation of NC and pretraining using NC require groundtruth relative poses during training, which is challenging to scale to complex tasks with multiple or deformable objects.

## ACKNOWLEDGMENTS

We would like to express our sincere appreciation to X.Y. Han for sharing valuable related works on Neural Collapse, and to Cheng Chi and Yilun Du for their insightful discussions on Diffusion Policy. We thank Yifeng Zhu, Shucheng Kang, and Yulin Li for their discussion and assistance with the Franka Panda robot arm setup. Additionally, Haocheng Yin acknowledges the Swiss-European Mobility Programme (SEMP) for their financial support, which partially covered travel expenses from ETH Zürich to Harvard University and living costs during the stay.

We are additionally grateful to the anonymous reviewers for their constructive feedback and valuable suggestions that have helped improve the clarity and technical depth of this paper. Their detailed comments have significantly strengthened our experimental analysis and theoretical discussions.

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

## A    LUNAR LANDER EXPERIMENT DETAILS

**Setup**    The Lunar Lander task is an environment in OpenAI Gym (Brockman, 2016), whose control space is discrete with 4 actions:

- 0: do nothing
- 1: fire left orientation engine
- 2: fire main engine
- 3: fire right orientation engine.

To collect an expert demonstration dataset on the lunar lander task, we trained an expert reinforcement learning algorithm —Proximal Policy Optimization (PPO) (Schulman et al., 2017)— on the LunarLander-v2 gym environment. We applied the default optimal network architecture and training hyperparameters from the popular RL training library – rl_zoo3 (Raffin, 2020) to train the PPO. Then we used the trained PPO policy to collect 500 demos as our expert training dataset.

**Training details**    After collecting expert demonstrations, we train an imitation learning model. We use a ResNet18 model as the vision encoder and use a MLP model to map the 512 dimensional latent embedding into 64 dimensional. We use a sequence of 4 input images as observation input and predict the next action. The action decoder is structured as another MLP network with 6 layers and maps the image latent embeddings to predicted action, then calculating the cross-entropy loss with the ground truth discrete actions. We train the model for 600 epochs.

## B    $k$-MEANS CLUSTERING OF EXPERT ACTIONS

Using the 500 expert demonstrations collected as described in §3.2, we attempt to perform a $k$-means clustering of the output actions directly to classify the training samples.

To decide what is the optimal number of clusters for $k$-means clustering, we follow the popular Silhouette analysis (Rousseeuw, 1987). Fig. 11 shows the average Silhouette score when varying the number of clusters, and Fig. 12 shows the Silhouette analysis for selected values of $k$. Based on the Silhouette analysis, we choose the optimal $k = 5$.

Fig. 13 then plots the three NC metrics calculated according to the classes determined from $k$-means clustering. We observe that although the STDAngle and STDNorm metrics decrease, the CDNV metric keeps increasing, providing a negative signal for neural collapse.

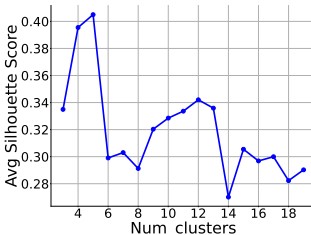

Figure 11: Average Silhouette score of $k$-means clustering of the output action with respect to number of clusters ($k$ value) from 3 to 19. The silhouette score is a metric used to evaluate the quality of clustering in $k$-Means clustering. The optimal $k$ value is 5, having the highest average Silhouette score.

## C    ACTION-BASED CLASSIFICATION

In addition to the Goal-based classification mentioned in §3.1, we introduce another possible classification method.

**Action-based classification**    Given a sample $s_t$ as in (8), we look at the images induced by the action sequence $(u_t, \ldots, u_{t+H-1})$, i.e., $(I_t, I_{t+1}, \ldots, I_{t+H})$. Importantly, for planar pushing, it

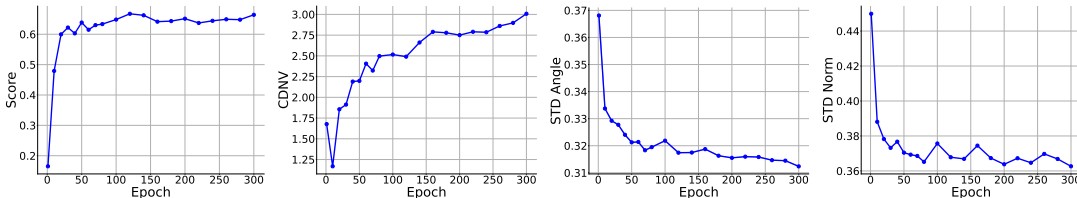

Figure 12: Silhouette analysis for $k$-Means clustering on sample data with number of clusters = 3, 5, 8, 12, 16 (from left to right). The red dotted vertical line represents the average Silhouette score while scores of samples in different clusters are represented with different colors.

Figure 13: $k$-Means clustering of the output actions does not lead to observation of neural collapse, even using the optimal $k$ value ($k = 5$). Data obtained from training a (ResNet+DM) model on 500 expert demonstrations described in §3.2.1.

is possible that the expert did not finish her action plan at time $t + H$, which can be detected if the pusher is still in contact with the object. In those cases, we enlarge $H$ until the pusher breaks contact with the object, which forms a full episode of expert plan. Fig. 14 right illustrates such an episode, with the object transparency diminishing as time progresses. In a full episode, we compute the relative pose of the object at time $t + H$ with respect to the object at time $t$, which consists of a triplet $(x, y, \theta)$ visualized in Fig. 14 right (Planar Pushing) and Fig. 15 right (Block Stacking).

The two classification strategies are correlated but different. Both of them measure the goal of the expert, but the first strategy measures the *long-term* goal while the second strategy measures the *short-term* goal (in an episode).

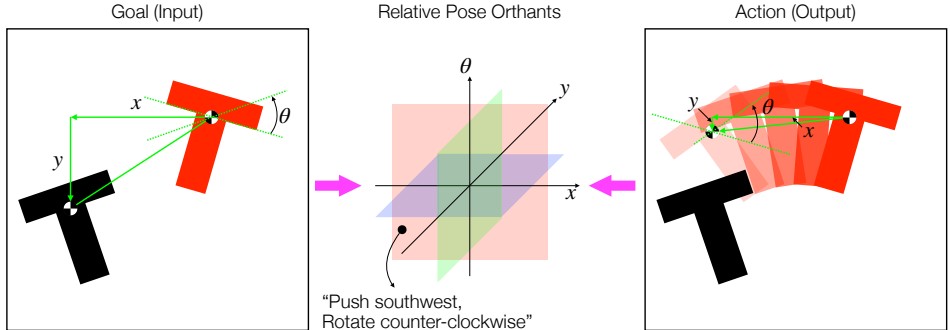

Figure 14: Control-oriented classification for Planar Pushing. Left: goal-based; Right: action-based.

## D   BENEFITS OF IMAGENET-PRETRAINED RESNET

On the planar pushing task, we investigate how much benefits does ImageNet pretraining bring to training a vision-based control pipeline.

**Push-T with clean background**   We compare the performance of four pipelines on the push-T task with a clean background (shown in Fig. 2). Table 2 second column shows the results.

In the table, "Frozen" indicates the ResNet weights are fixed and not finetuned in behavior cloning, while "Finetuned" indicates the weights are end-to-end finetuned from expert demonstrations. "Pre-

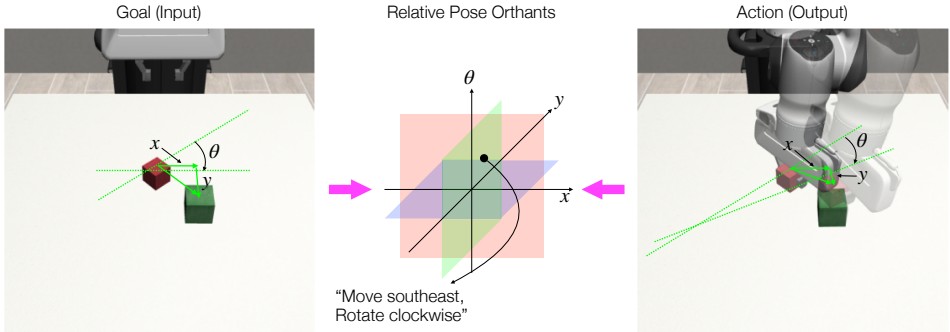

Figure 15: Control-oriented classification for Block Stacking. Left: goal-based; Right: action-based.

trained" indicates the ResNet weights are initialized from pretraining on ImageNet, and "Random" indicates a random set of initial weights. In all cases, the action decoder, i.e., the diffusion model (DM) is not frozen and trained end-to-end.

Clearly, the ImageNet-pretrained ResNet does not show benefits either when frozen or finetuned, compared to a set of random weights.

**Push-T with complex background**   We replace the clean background of push-T with a messy table-top manipulation background adapted from the well-known LineMod dataset Hinterstoisser et al. (2013). An example image of the generated expert demonstrations can be found in Fig. 19. Table 2 third column shows the results. The pipelines are the same as before.

We make two observations:

- When the ResNet is frozen, regardless of whether pretrained from ImageNet or randomly initialized, the performance is very low. This shows that, if finetuning is not allowed, pretraining on ImageNet does not help.
- However, when finetuning is allowed, pretraining on ImageNet significantly increases the performance. This shows the rich priors learned from ImageNet indeed helps transfer learning for control tasks, when the scenes are visually complex.

Combining the results above, we draw the following conclusions.

- In control tasks, if ResNet is not allowed to be finetuned, then neither pretraining on ImageNet nor random initialization is sufficient to learn a good policy for the control task.
- When ResNet is allowed to be finetuned:
  - If the scene is visually simple (such as a clean push-T), then pretraining on ImageNet does not show clear advantages over a random initialization.
  - If the scene is visually complex (with a complex background), then pretraining on ImageNet significantly benefits transfer learning.

| | Performance on clean push-T (%) | Performance on visually complex push-T (%) |
|---|---|---|
| Frozen Pretrained ResNet + DM | 30.17 | 32.90 |
| Frozen Random ResNet + DM | 29.95 | 32.22 |
| Finetuned Pretrained ResNet + DM | 62.83 | 56.10 |
| Finetuned Random ResNet + DM | 63.66 | 46.67 |

Table 2: Investigation of the benefits of ImageNet Pretraining on planar pushing.

# E   NEURAL RE-COLLAPSE DURING FINETUNING

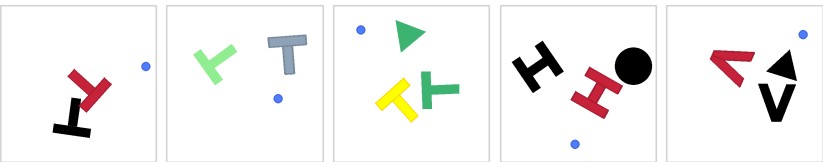

Figure 16: Continual training of a (ResNet+DM) image-based control pipeline on five different planar pushing domains with changes in object color, shape, and visual distractors.

We collect 500 expert demonstrations in each of the five planar pushing domains shown in Fig. 16. We then continuously train a (ResNet+DM) model across five domains, each with 300 epochs. Fig. 17 shows the performance score and three NC metrics (according to the goal-based classification), where blue curves correspond to data in the target (new) domain and yellow curves correspond to data in the source (old) domain. We make three observations. (i) During retraining on the target domain, performance increases in the target domain but drops in the source domain. (ii) The NC metrics consistently decrease when retraining on the target domain (and increase or plateau on the source domain), suggesting the visual representation space is "re-clustered". (iii) Contradictory to recent literature that observe *loss of plasticity* in continual learning (Dohare et al., 2024; Muppidi et al., 2024), there is no clear evidence the (ResNet+DM) model loses its ability to learn new tasks during continual finetuning (in every new domain, the model reaches around 70% performance). Results using the action-based classification are similar and presented in Fig. 18.

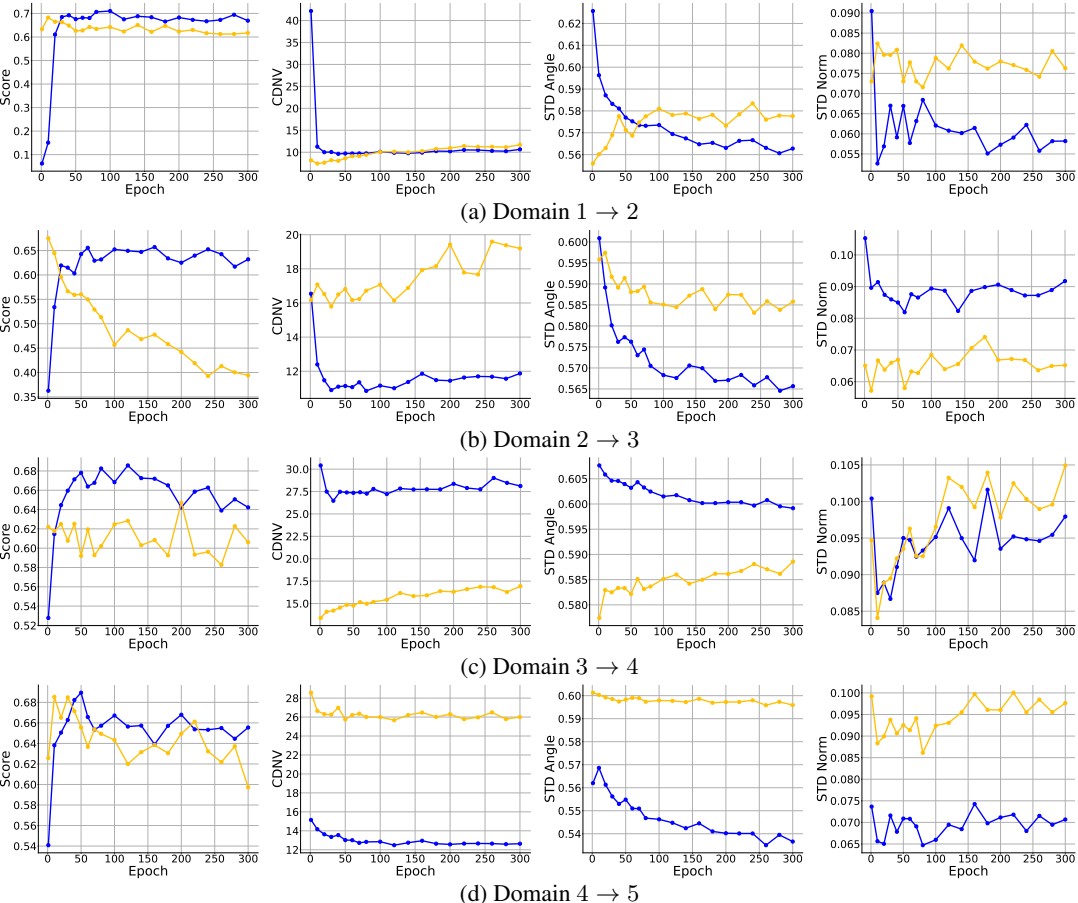

Figure 17: Neural re-collapse in the visual representation space during continual learning across different domains. Blue curves represent data in the target (new) domain, and yellow curves represent data in the source (old) domain. All results use the goal-based classification. Results using the action-based classification are similar and shown in Fig. 18.

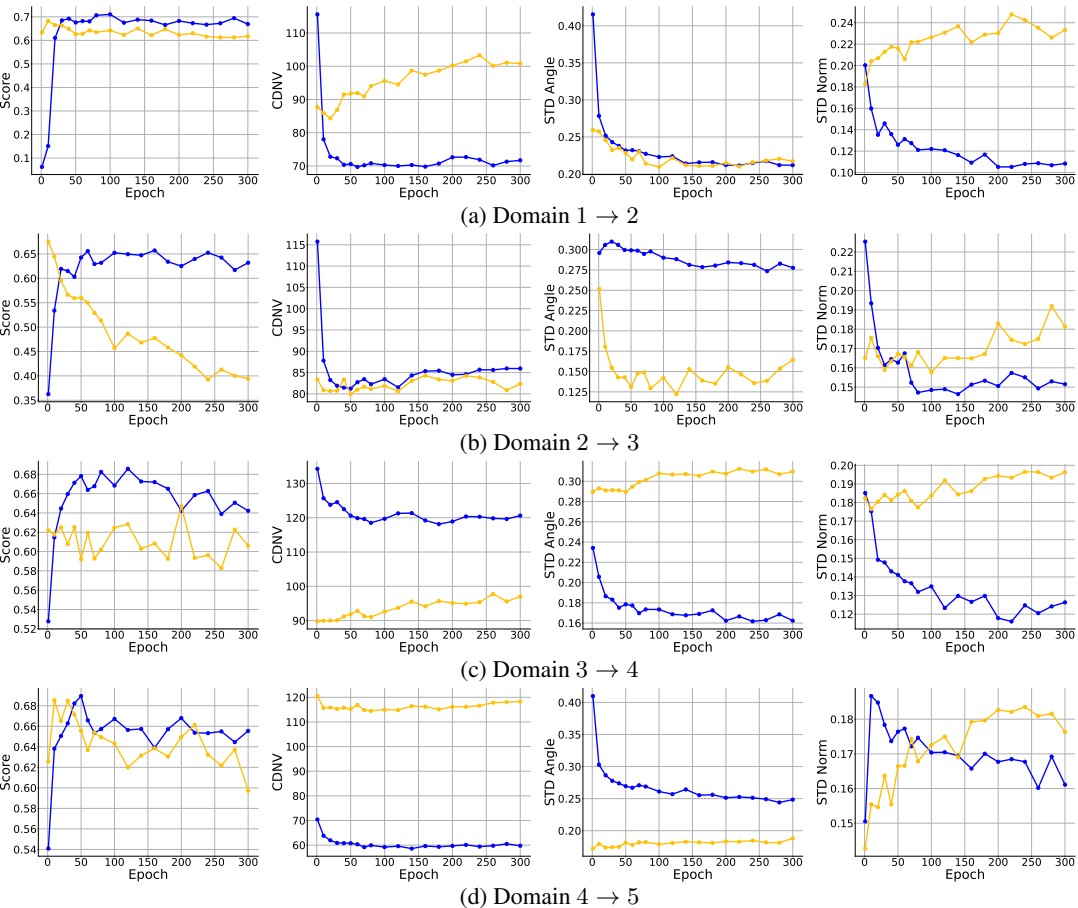

Figure 18: Neural re-collapse in the visual representation space during continual retraining across different domains. All results use the action-based classification.

# F DETAILS OF VISUAL REPRESENTATION PRETRAINING

**Training details** In Table 1, for both the baseline and our method, the network architecture is exactly the same – We used ResNet as the vision encoder and diffusion model (DM) as the action decoder. Baseline trains the entire pipeline end-to-end with expert demonstrations using action noise L2 loss supervision for 100 epochs, which ensures the loss fully converged.

Our method is different from the baseline only in terms of training strategy. We firstly pretrain the same ResNet using NC regularization (i.e., minimize the NC metrics and encourage the visual features to cluster according to control-oriented labels) for 50 epochs, and then end-to-end train pretrained ResNet+DM pipeline from expert demonstrations for another 50 epochs. During NC regularized pretraining, because all the data samples cannot be fit into one batch due to GPU limitation, we use the largest possible batch size and divide the randomly shuffled dataset into 4 batches in every pretraining epoch. We calculate the NC regularization for each batch following this NC loss: $0.1 \times \text{CDNV} + 10 \times (\text{STDNorm} + \text{STDAngle})$.

We want to emphasize that our pipeline with control-oriented pretraining does **NOT** use any extra information compared to the baseline during **TEST** time. At test time, our pipeline is still a pure vision-based control pipeline.

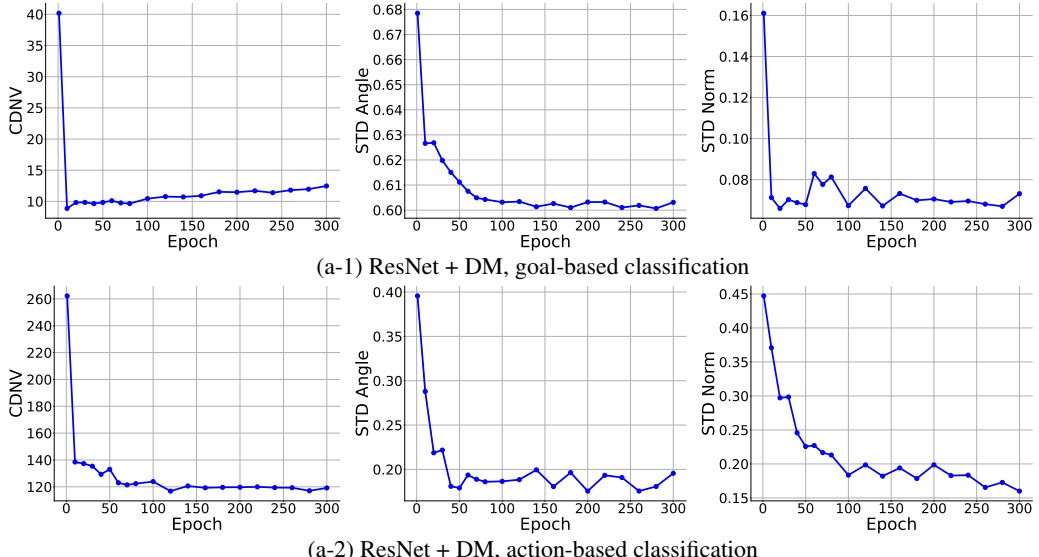

(a-1) ResNet + DM, goal-based classification

(a-2) ResNet + DM, action-based classification

Figure 20: Prevalent emergence of neural collapse in the visual representation space for **complex background push-T**. (a) Three NC metrics w.r.t. training epochs using ResNet as the vision encoder and diffusion model (DM) as the action decoder.

# G    SUPPLEMENTARY EXPERIMENTAL RESULTS

## G.1    SIMULATED PLANAR PUSHING WITH VISUALLY COMPLEX BACKGROUND

**Setup**    Similar to the planar pushing setup in main text, we collect $N = 500$ expert demonstration on a push-T setup with a messy table-top manipulation background adapted from the well-known LineMod dataset (Hinterstoisser et al., 2013). An example of this dataset can be seen in Fig 19. At each round, the object and target positions are randomly initialized and the same human expert controls the pusher to push the object into alignment with the target position. We define our primary evaluation metric as the ratio of the overlapping area between the object and the target position to the total area of the target position.

**Results**    We used 500 demonstrations to train an end-to-end pipeline using ResNet as the vision encoder (pretrained from ImageNet) and Diffusion Model as the action decoder. The test-time performance for a well-trained model is $56.1\%$. Fig. 20 plots the three NC evaluation metrics w.r.t. training epochs with both goal-based and action-based classification. We observe consistent decrease of three NC metrics as training progresses, suggesting the prevalent emergence of a law of clustering that is similar to NC, even though in a task with many visual distractions.

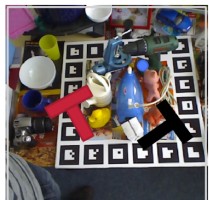

Figure 19: Push-T in LineMod background.

## G.2    NEURAL COLLAPSE IN PLANAR PUSHING WITH ACTION-BASED CLASSIFICATION

Fig. 21 plots the three NC evaluation metrics w.r.t. training epochs for two different models (both with DM) with **action-based classification label** described in Appendix C. We also observe consistent decrease of three NC metrics as training progresses, suggesting the prevalent emergence of a law of clustering that is similar to NC. Notably, the CDNV metric in the case of using a goal-based classification as in Fig. 6 is significantly smaller than that of an action-based classification, suggesting that the goal-based classification may correspond to a stronger level of clustering.

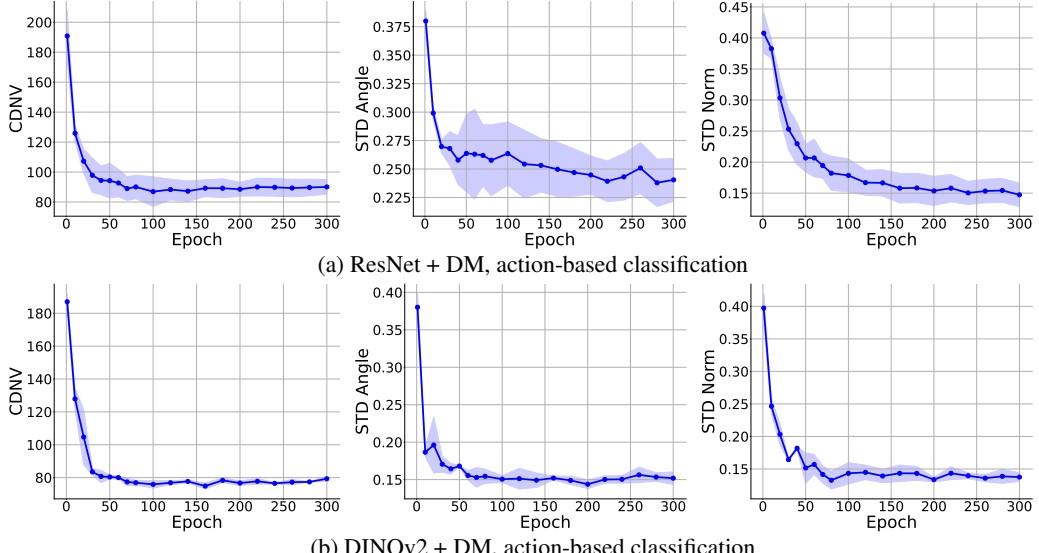

(a) ResNet + DM, action-based classification

(b) DINOv2 + DM, action-based classification

Figure 21: Prevalent emergence of neural collapse in the visual representation space for planar pushing. (a) Three NC metrics w.r.t. training epochs using ResNet as the vision encoder and diffusion model (DM) as the action decoder, with action-based classification label. (b) Three NC metrics w.r.t. training epochs using DINOv2 as the vision encoder and DM as the action decoder. All the plots show the mean and standard deviation (in shaded band) over three random seeds. The test scores are shown in Fig. 5(a)(b).

### G.3   Observation of Neural Collapse with LSTM

In the main text, we demonstrated emergence of neural collapse when using diffusion model as the action decoder. Here we provide evidence of similar neural collapse when using LSTM as the action decoder, as shown in Fig. 22.

### G.4   Finegrained Repos

In Remark 1 we stated that when increasing the number of classes used to divide the space of relative pose, neural collapse still holds. Here we provide more details.

Recall that the relative pose is a triplet $(x, y, \theta)$ containing a 2D translation and a 1D rotation. In the 8-class division, we simply divide each dimension $(x, y, \theta)$ into two bins $(-\infty, 0]$ and $[0, +\infty)$, leading to $2^3 = 8$ classes. To divide the space into more classes, we can divide each dimension into more bins. In particular, we try two extra options. First, we divide each dimension into 4 bins, leading to 64 classes. Second, we divide each dimension into 6 bins, leading to 216 classes. Fig. 23 compares the NC metrics under 8, 64, and 216 classes using the (ResNet+DM) model and the goal-based classification strategy. We can clearly observe that when the number of classes is increased, neural collapse still emerges. However, it appears that the CDNV metric can "bounce back" when the number of classes is increased. Similar observations hold for the (DINOv2+DM) model and shown in Fig. 24.

### G.5   Weak Clustering

In Fig. 8 we showed weak clustering when training using 100 demonstrations according to the goal-based classification. Here Fig. 25 shows weak clustering according to the action-based classification.

### G.6   Visual Representation Pretraining with NC for DINOv2

In addition to pretrain a ResNet vision encoder with NC regularization as in Table 1, we also present results of using DINOv2 as vision encoder and pretrain it by minimizing the NC metrics in Table 3.

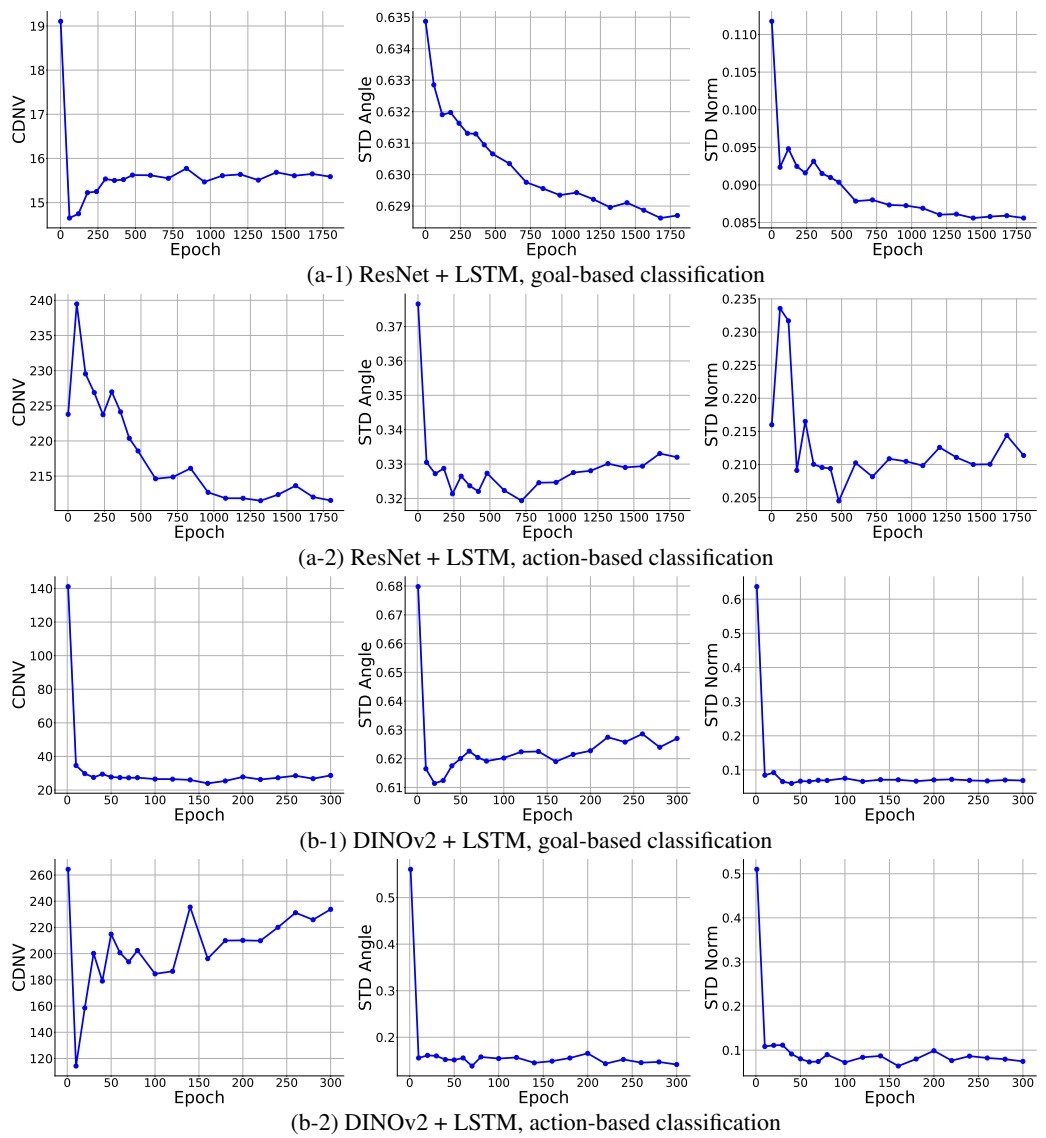

Figure 22: Prevalent emergence of neural collapse in the visual representation space. (a) Three NC metrics w.r.t. training epochs using ResNet as the vision encoder and LSTM as the action decoder. (b) Three NC metrics w.r.t. training epochs using DINOv2 as the vision encoder and LSTM as the action decoder. The test scores are shown in Fig. 5 (c)(d). Only one seed of results are shown because the test performance is poor when using LSTM as the action decoder. NC results when using diffusion model as the action decoder are shown in Fig. 6 and Fig. 21.

Clearly, we observe that control-oriented pretraining improves test-time performance, regardless of whether ResNet or DINOv2 are used.

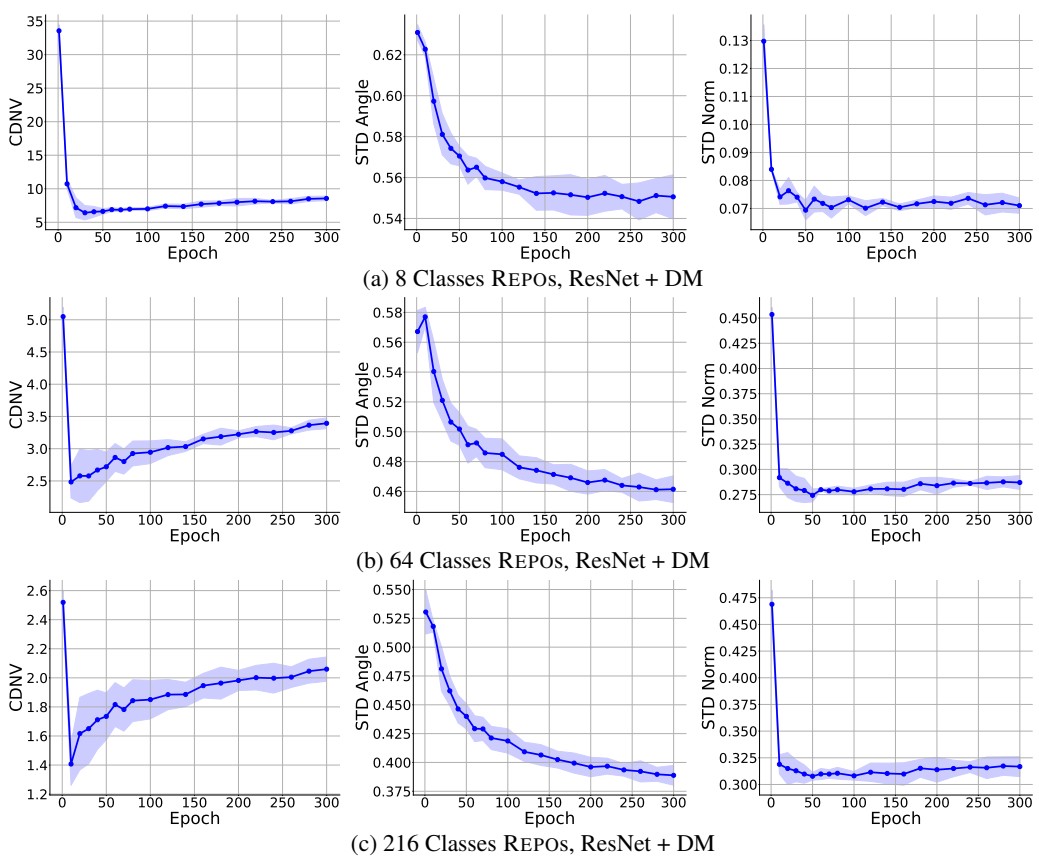

(a) 8 Classes REPOs, ResNet + DM

(b) 64 Classes REPOs, ResNet + DM

(c) 216 Classes REPOs, ResNet + DM

Figure 23: NC metrics of finegrainded REPOs for ResNet + DM model. (a-c) represent the three NC metrics w.r.t. training epochs using ResNet as the vision encoder and diffusion model (DM) as the action decoder, respectively for 8, 64, 216 classes of REPOs.

| Pushing tasks | | | | |
|---|---|---|---|---|
| Baseline(w/ DINOv2) | 0.1895 | 0.3747 | 0.2463 | 0.1942 |
| Control-oriented pretraining (w/ DINOv2) | 0.2110 | 0.4602 | 0.2466 | 0.2468 |

Table 3: Pretraining a DINOv2 vision encoder by minimizing the NC metrics also improves test-time model performance on four domains corresponding to four letters in the word "ROBOT".

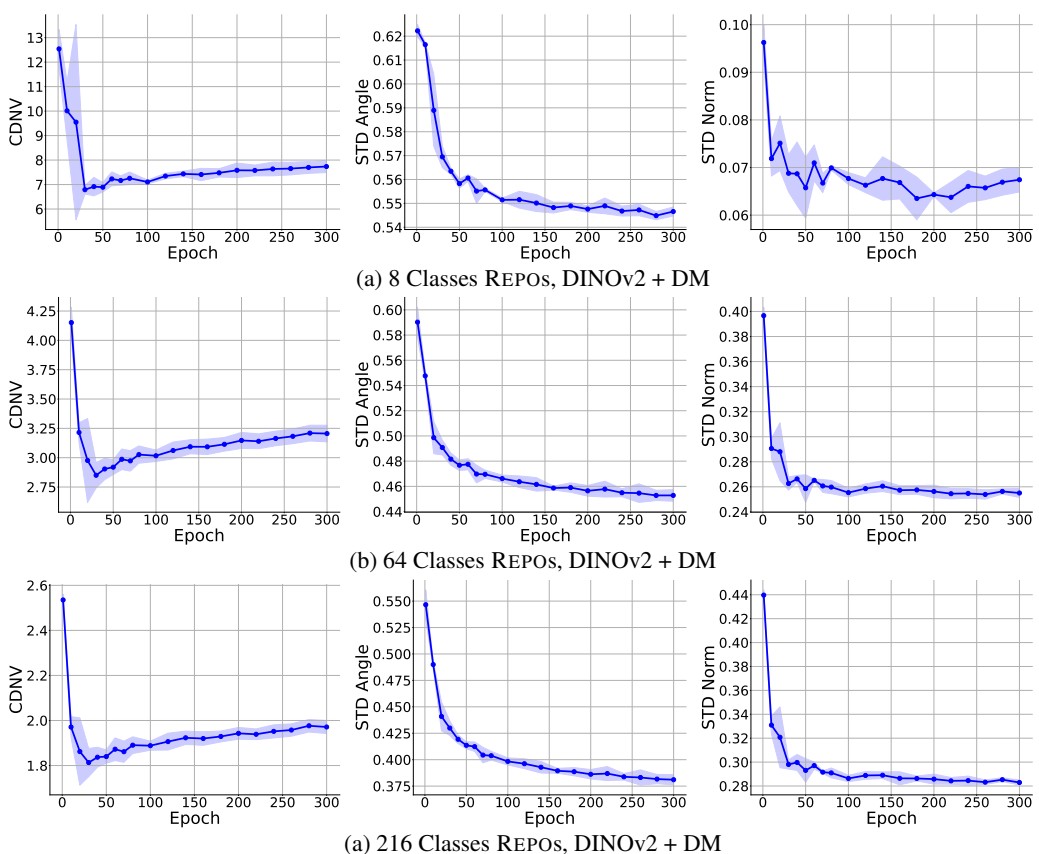

Figure 24: NC metrics of finegrainded REPOs for DINOv2 + DM model. (a-c) represent the three NC metrics w.r.t. training epochs using DINOv2 as the vision encoder and diffusion model (DM) as the action decoder, respectively for 8, 64, 216 classes of REPOs.

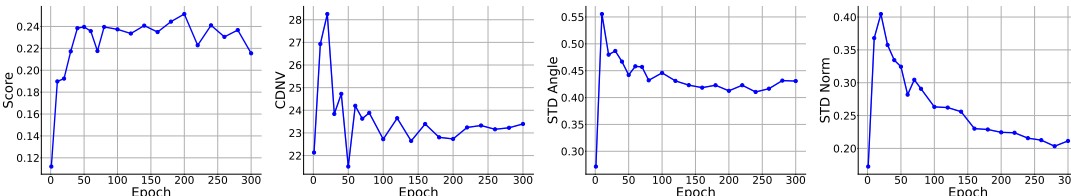

Figure 25: Weak neural collapse under 100 demonstrations. NC metrics are calculated using action-based classification.

## H    REAL-WORLD EXPERIMENTAL RESULTS

Here we provide full results for real-world experiments.

- **100 demonstrations on clean background**  In the main text we showed results for 1 test. Fig. 28 and 29 show the results for all 10 tests. As seen, the NC-pretrained policy succeeded 8 out of 10, but the baseline policy succeeded 5 out of 10.
- **50 demonstrations on clean background**  In addition, we also trained two policies using only 50 demonstrations. Fig. 26 and 27 show the results for all 10 tests. The NC-pretrained policy succeeded 4 out of 10, but the baseline policy succeeded 2 out of 10.
- **200 demonstrations on visually complex background**  In the main text we showed results for 1 test. Fig. 30 and 31 show the results for all 10 tests. As seen, the NC-pretrained policy succeeded 9 out of 10, but the baseline policy succeeded 6 out of 10.

For video recordings of the test results, please consult the supplementary material.

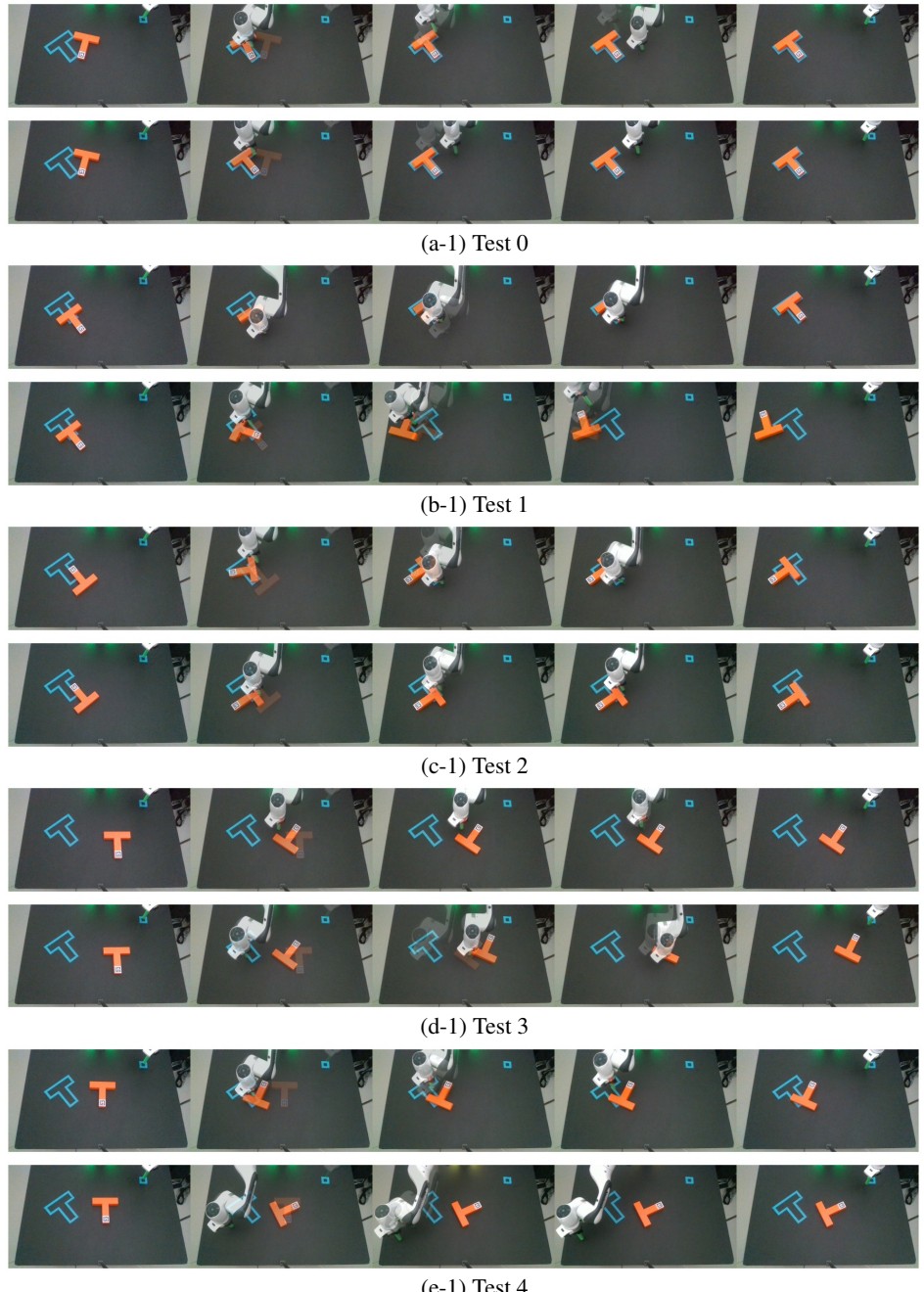

(a-1) Test 0

(b-1) Test 1

(c-1) Test 2

(d-1) Test 3

(e-1) Test 4

Figure 26: Real-world evaluation on **clean background** for test 0 to test 4. In every test, top row shows trajectories of NC-pretrained model and bottom row shows trajectories of baseline. Both models are trained under **50 training demonstrations**. The first and last column show the initial and final position of the object, respectively. The middle three columns show overlaid trajectories generated by the policies.

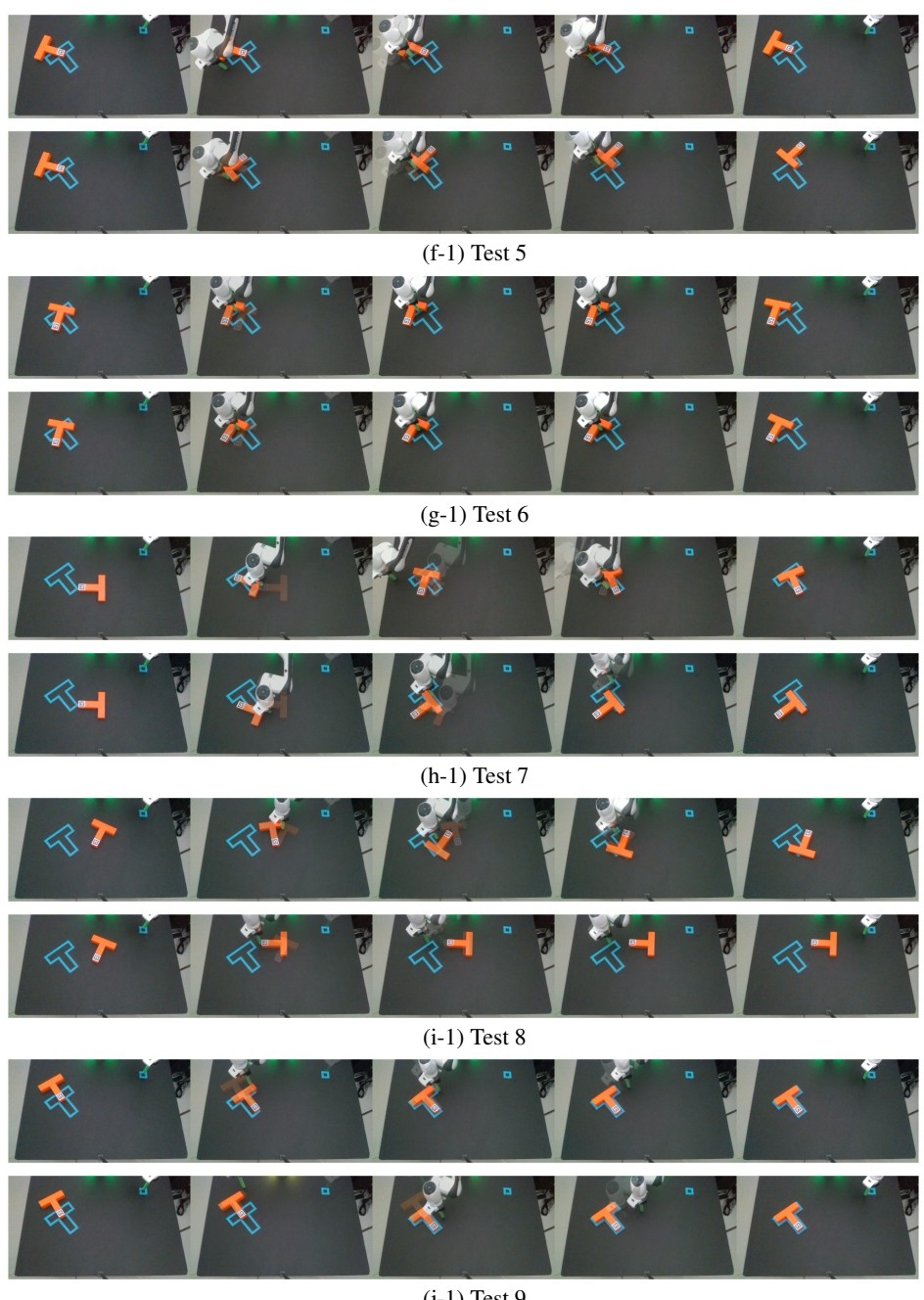

(f-1) Test 5

(g-1) Test 6

(h-1) Test 7

(i-1) Test 8

(j-1) Test 9

Figure 27: Real-world evaluation on **clean background** for test 5 to test 9. In every test, top row shows trajectories of NC-pretrained model and bottom row shows trajectories of baseline. Both models are trained under **50 training demonstrations**. The first and last column show the initial and final position of the object, respectively. The middle three columns show overlaid trajectories generated by the policies.

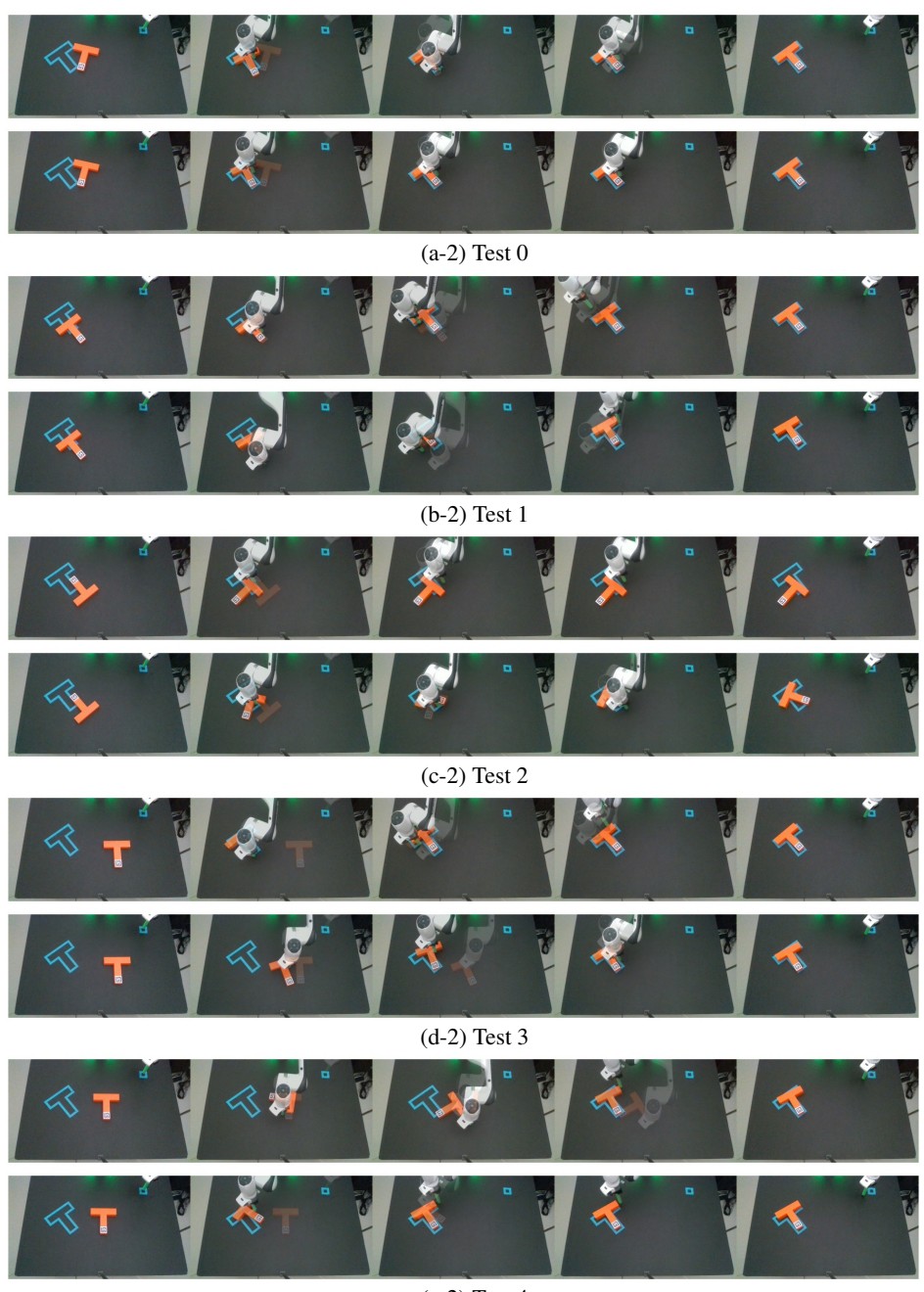

(a-2) Test 0

(b-2) Test 1

(c-2) Test 2

(d-2) Test 3

(e-2) Test 4

Figure 28: Real-world evaluation on **clean background** for test 0 to test 4. In every test, top row shows trajectories of NC-pretrained model and bottom row shows trajectories of baseline. Both models are trained under **100 training demonstrations**. The first and last column show the initial and final position of the object, respectively. The middle three columns show overlaid trajectories generated by the policies.

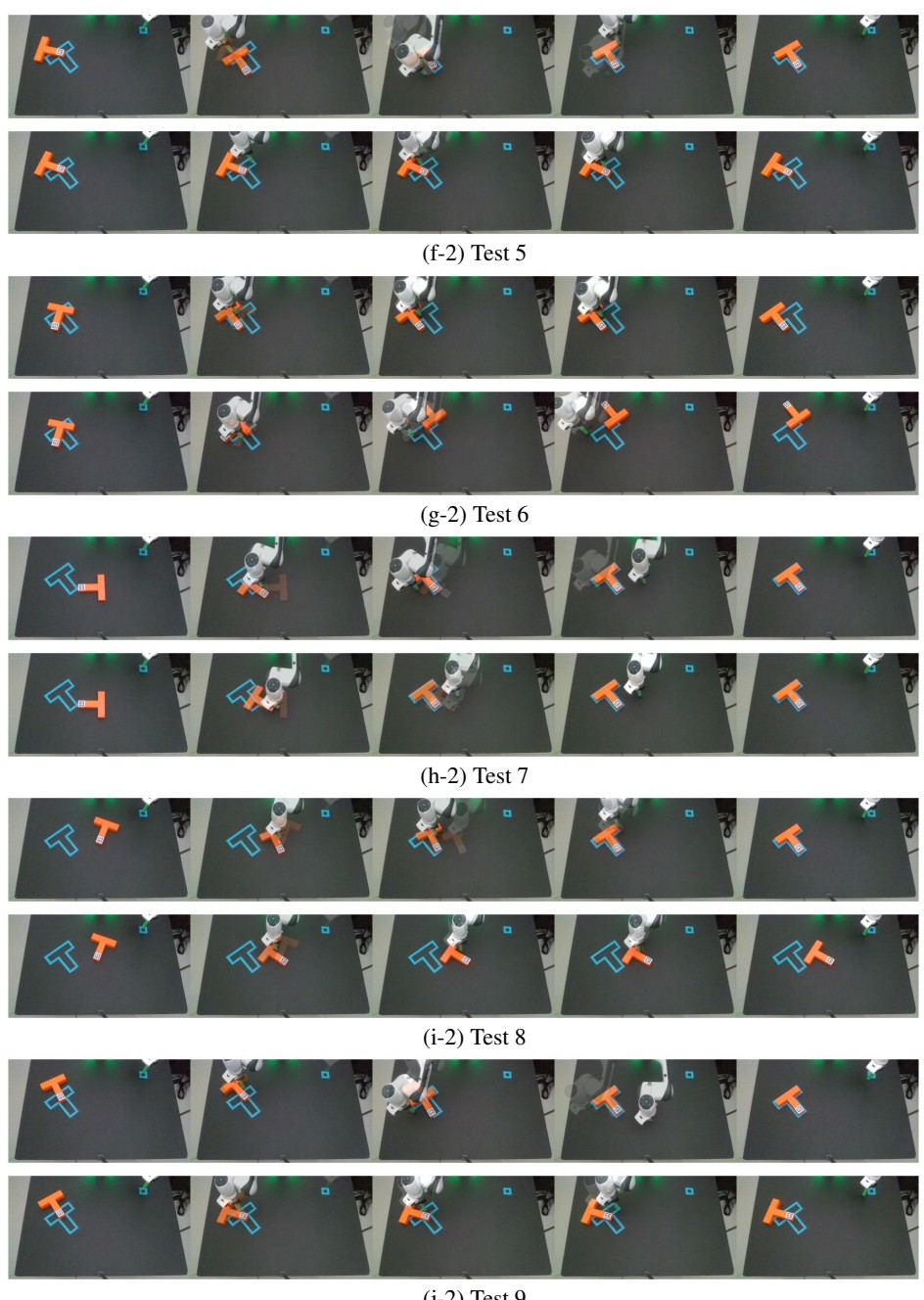

(f-2) Test 5

(g-2) Test 6

(h-2) Test 7

(i-2) Test 8

(j-2) Test 9

Figure 29: Real-world evaluation on **clean background** for test 5 to test 9. In every test, top row shows trajectories of NC-pretrained model and bottom row shows trajectories of baseline. Both models are trained under **100 training demonstrations**. The first and last column show the initial and final position of the object, respectively. The middle three columns show overlaid trajectories generated by the policies.

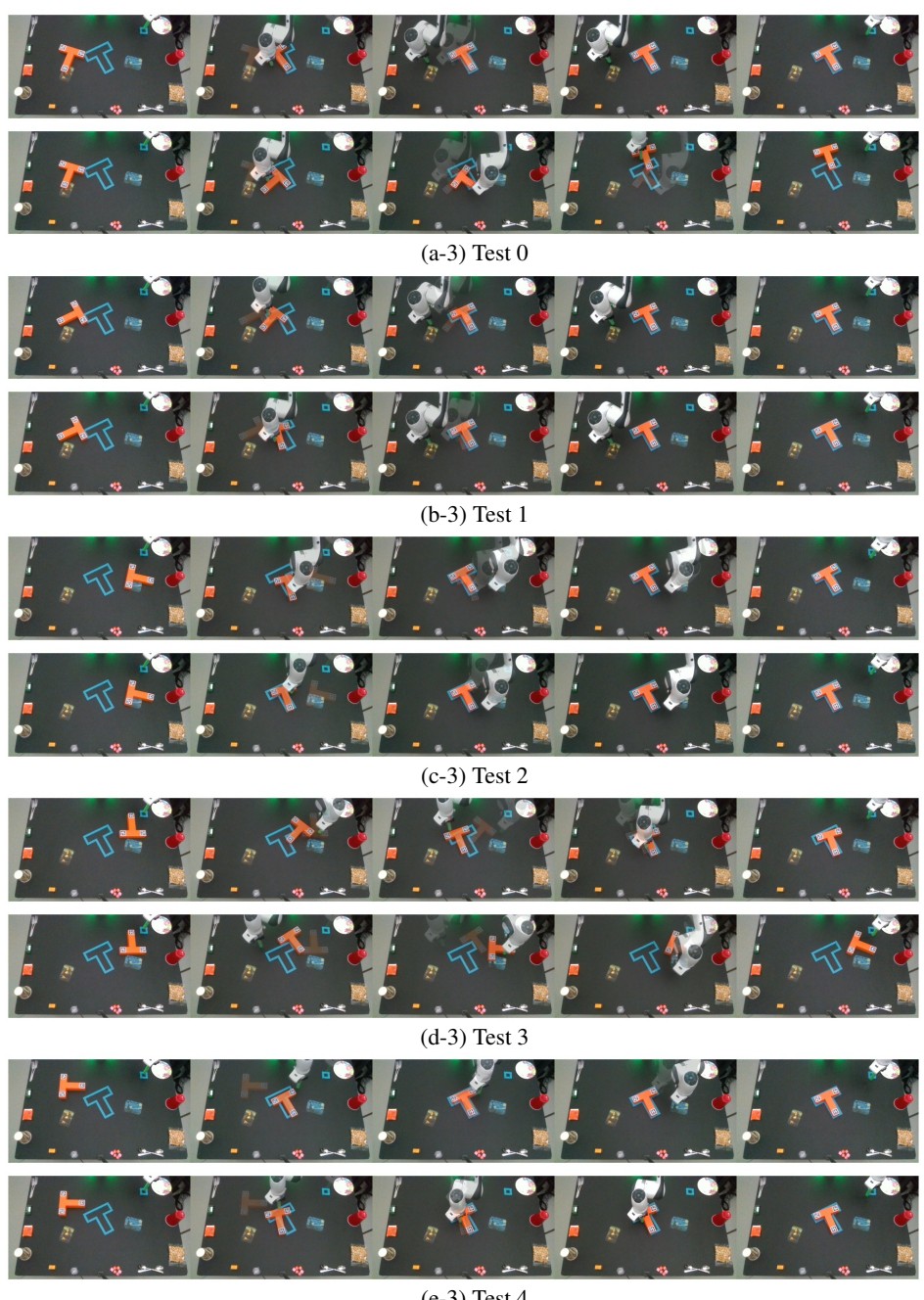

(a-3) Test 0

(b-3) Test 1

(c-3) Test 2

(d-3) Test 3

(e-3) Test 4

Figure 30: Real-world evaluation on **visually complex background** for test 0 to test 4. In every test, top row shows trajectories of NC-pretrained model and bottom row shows trajectories of baseline. Both models are trained under **200 training demonstrations**. The first and last column show the initial and final position of the object, respectively. The middle three columns show overlaid trajectories generated by the policies.

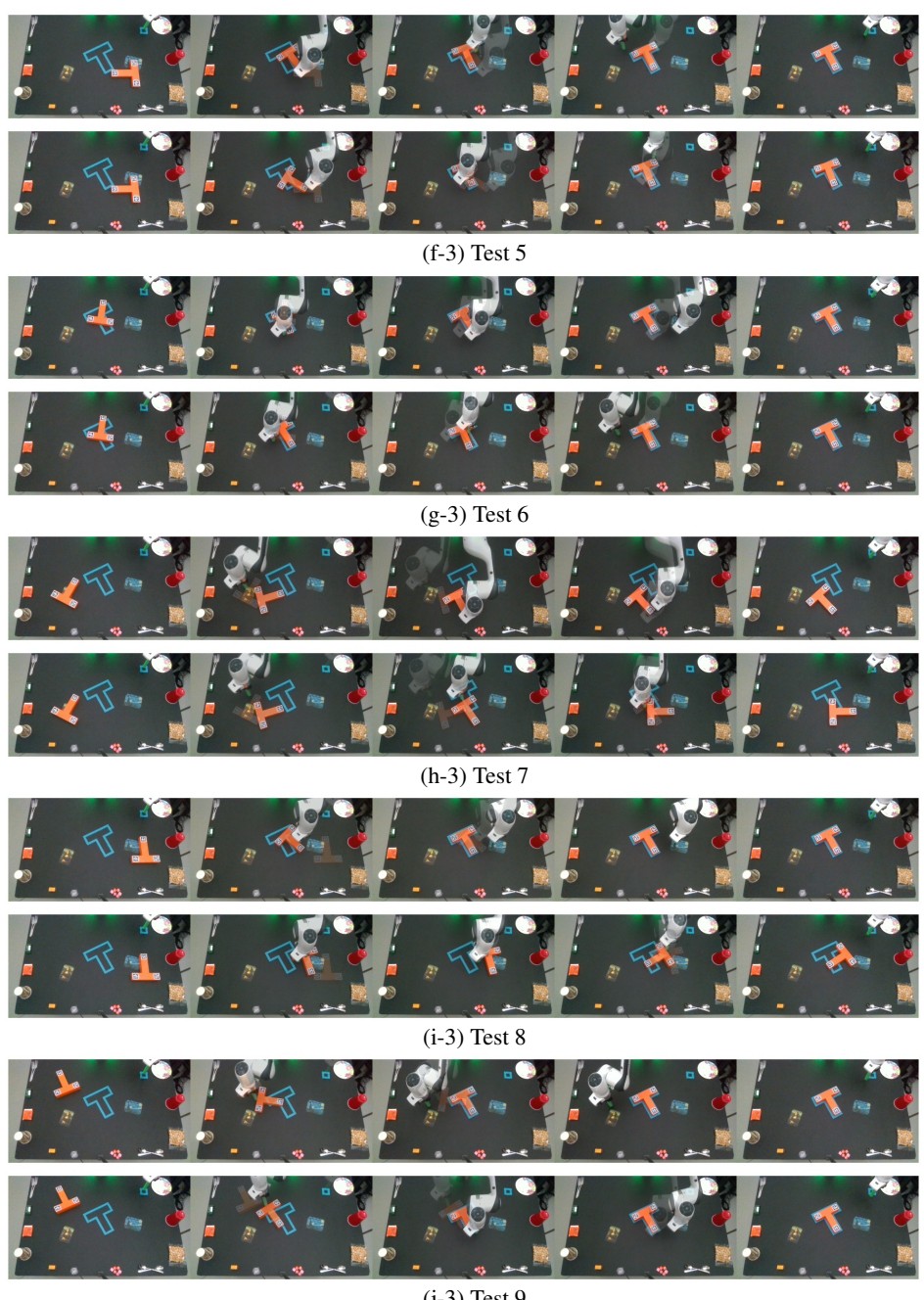

(f-3) Test 5

(g-3) Test 6

(h-3) Test 7

(i-3) Test 8

(j-3) Test 9

Figure 31: Real-world evaluation on **visually complex background** for test 5 to test 9. In every test, top row shows trajectories of NC-pretrained model and bottom row shows trajectories of baseline. Both models are trained under **200 training demonstrations**. The first and last column show the initial and final position of the object, respectively. The middle three columns show overlaid trajectories generated by the policies.

## I    RELATED WORK

**Neural Collapse**    The phenomenon of neural collapse (NC) is firstly introduced in Papyan et al. (2020), about an elegant geometric structure of the last-layer feature and classifier for a well-trained model in classification tasks. While the initial NC is observed for model trained with cross entrophy (CE) loss, Han et al. (2021) demonstrated NC also holds when using the Mean Squared Error (MSE) loss for training.  With the simplified assumption that only considers the last-layer optimization, neural collapse is proved to be the phenomenon occurring when training achieves global optimality, trained with CE loss function (E & Wojtowytsch, 2020; Zhu et al., 2021), and MSE loss function (Zhou et al., 2022; Tirer & Bruna, 2022).  Later, NC is studied in more diverse situations, e.g., NC in imbalanced training (Fang et al., 2021; Hong & Ling, 2023; Dang et al., 2024), NC through intermediate layers (Rangamani et al., 2023; Parker et al., 2023), NC in transfer learning (Galanti et al., 2021; 2022), and relationship between NC and optimization methods (Xu et al., 2023; Rangamani & Banburski-Fahey, 2022).  Since NC is a critical indicator of model robustness, it is also studied as a tool to improve the model (Bonifazi et al., 2024; Liu et al., 2023; Yang et al., 2022).  Recently, neural collapse is studied in more complex applications, like large language model training (Wu & Papyan, 2024; Jiang et al., 2023).  These works study neural collapse in the classification tasks and gain insightful results about the relationship between NC and training deep neural networks. In this work, we study NC of the visual representation space in an image-based control pipeline and show that control-oriented classification plays a key role in identifying NC.

**Behavior Cloning**    Behavior Cloning (BC) is a fundamental approach in Imitation Learning, where an agent learns to mimic expert demonstrations (Argall et al., 2009; Hussein et al., 2017). It typically involves collecting expert demonstrations and training a supervised learning model to reproduce the demonstrated behavior.  While conceptually simple, BC has proven effective in many scenarios (Muller et al., 2005), particularly when combined with data augmentation techniques (Laskey et al., 2017; Peng et al., 2018) or iterative refinement (Ross & Bagnell, 2010; Ho & Ermon, 2016).  This method has been widely applied across various robotic tasks including arm manipulation (Sharma et al., 2018; Rajeswaran et al., 2018), autonomous driving (Bojarski, 2016; Codevilla et al., 2018), and game playing (Silver et al., 2016; Vinyals et al., 2019).  However, BC faces challenges such as compounding errors (Ross et al., 2011) and struggles with long-horizon tasks (Sun et al., 2017).  Recent advancements have focused on addressing these limitations through improved data collection strategies (Mandlekar et al., 2018), robust loss functions  (Reddy et al., 2019; Brown et al., 2019), and integration with other learning paradigms (Rajeswaran et al., 2018; Zhu et al., 2020).  In this work, we study BC from a representation perspective, and demonstrated that representation regularization, using neural collapse, can effectively improve the performance of BC in the low-data regime.

**Diffusion Policy**    The advent of diffusion models in imitation learning, exemplified by Diffusion Policy (Chi et al., 2023), has introduced a groundbreaking approach to behavior cloning for robot control. This method leverages the principles of denoising diffusion probabilistic models (Ho et al., 2020) to learn a policy that progressively refines random noise into expert-like actions.  By doing so, it offers a powerful framework for modeling complex, multi-modal action distributions, which is particularly beneficial in robotics tasks involving high-dimensional action spaces (Janner et al., 2022).  Diffusion policy has demonstrated impressive results in challenging robotics tasks, effectively addressing longstanding issues in traditional behavior cloning approaches such as distribution shift and long-horizon planning (Wang et al., 2022).  As the field progresses, diffusion policy stands as a promising direction for enhancing the capabilities and robustness of imitation learning in complex robotic systems. In this work, we show that probing the geometry of the visual representation space helps understand and improve diffusion policy.

