# OpenReview forum: "Control-oriented Clustering of Visual Latent Representation"
_ICLR.cc/2025/Conference — ICLR 2025 Spotlight_

### Official Review · Reviewer_NdZb · 2024-11-03

**Soundness:** 4
**Presentation:** 4
**Contribution:** 4
**Rating:** 8
**Confidence:** 4

**Summary:**

This paper investigates the geometry of the visual representation space in an image-based control pipeline trained through behavior cloning, focusing specifically on planar pushing tasks. It explores whether a phenomenon similar to neural collapse (NC), originally observed in image classification, can emerge in a regression context such as control tasks. The authors introduce "control-oriented" classes based on the relative poses of objects and targets, or relative poses of objects at different time steps, leading to the following key findings:
1. Control-oriented clustering emerges in the visual representation space, with latent features forming clusters according to control-relevant classes.
2. Using NC as a pretraining strategy for the vision encoder improves test-time performance, particularly when training data is limited.
3. Real-world experiments validate the effectiveness of NC pretraining in enhancing control performance.

**Strengths:**

1. Novel and interesting results: The paper presents intriguing and slightly surprising results, showing that control-oriented clustering in visual representations emerges even though the planar pushing task involves continuous control signals. Additionally, it demonstrates that pretraining the visual encoder improves test-time performance when training with limited expert demonstrations.
2. Excellent presentation and writing: The presentation of this paper is clear and engaging. The authors begin with a motivating toy example that introduces the phenomenon of NC, then proceed to answer two key questions: (a) Does NC occur in the visual representation space? and (b) Is the extent of NC related to model performance? The arguments are supported with convincing evidence, and the writing is very easy to follow.
3. Real-World Validation: The paper includes real-world experiments on vision-based planar pushing to show the effectiveness of NC pretraining

Overall, this paper presents novel and interesting results, with clear presentation and strong supporting evidence. This paper also opens up lots of future research directions in visual representation learning for robotics control, which will greatly benefit the community. This paper is clearly above the acceptance bar for ICLR.

**Weaknesses:**

1. Scalability of the proposed approach for more complex tasks: Leveraging NC for control requires manually specified class definitions. In the planar pushing task, these classes are defined using the ground-truth relative pose of the target position with respect to the object, or the relative pose of the object at time t+H with respect to the object at time t. These classes are easy to define because the T-pushing task involves a single object, and its pose can be easily estimated using an AprilTag attached at the bottom of the T. However, the same class definition and pre-training strategy may not apply to more complex real-world tasks. In real-world robotic applications with diverse objects, accurate pose estimation can be challenging, particularly for deformable objects where defining a canonical pose is difficult. While accurate pose estimation is not the focus of this paper, it would be beneficial to discuss the potential limitations of the proposed class definitions in scaling to more complex tasks, or consider possible alternative class definitions.
2. Limited pretraining details and ablation studies: The paper lacks details about the visual pre-training, such as training datasets and training schedules. Moreover, there are no ablation studies provided on the relative importance of the two NC metrics in pretraining. Without these details, reproducing the results from the paper is challenging. Including such information would improve the reproducibility and robustness of the findings.

**Questions:**

Why does the clustering phenomenon emerge for the planar pushing task with a continuous control signal? The toy example has discrete states in the closed-form optimal policy, so one can see why the clustering phenomenon might arise. The planar pushing task presents a different scenario where this emergence seems less intuitive.

---

> ### Author Response · Authors · 2024-11-17
> **Response to Reviewer NdZb (I)**
>
> We thank the reviewer for the accurate and detailed summary of our paper and contributions.
>
> We next respond to the reviewer's concerns.
>
> ### **[A] "Scalability of the proposed approach for more complex tasks"**
>
> This is a very valid concern. Indeed in our own opinion, this is the biggest limitation of the current work, and we are actively thinking about approaches to address it.
>
> Below are some (perhaps immature) thoughts:
>
> - When there are multiple rigid-body objects in the scene, a potential solution is to use recent works that provide zero-shot pose estimation for many different objects, such as [FoundationPose](https://nvlabs.github.io/FoundationPose/). This will help annotate the groundtruth poses of different objects. Since the pose annotations are likely to be noisy or even contain outliers, two research problems are of interest: (a) does NC pretraining under **noisy labels** still improve performance of the end-to-end policy, and (b) how to remove the bad pose annotations. Of course, FoundationPose (and other recent learning-based methods) is not the only approach, and one can use many classical pose estimation methods based on feature matching and geometric optimization.
>
> - In the case of deformable objects, we are not sure. It seems to be a much more complicated problem due to the unclear definition of states. One potential way is to define several keypoints on the deformable object and use them to provide a low-dimensional representation of the object's state. One would perhaps need to look into the extensive literature of [deformable object manipulation](https://www.science.org/doi/10.1126/scirobotics.abd8803) to come up with more clever approaches.
>
> We will add a limitations and future work paragraph to our paper. Thanks for sharing these challenges!
>
>
>
> ---
>
>
> ### **[B] "Limited pretraining details and ablation studies"**
>
> We have added more experiments and ablation studies, please see our responses to other reviewers. Those include visually more complex scenes, block stacking experiments, and experiments on vision-based discrete-control problems.
>
> For the pretraining details, we provide them below and will add to our paper.
>
> We pretrain the vision encoder using all the training dataset (100 demos for experiments in Table 1). Because all the data samples cannot be fit into one batch due to GPU limitation, we use the largest possible batch size and divide the randomly shuffled dataset into 4 batches in every pretraining epoch. We calculate the NC regularization for each batch following this NC loss:
>
> $$0.1 \times \text{CDNV} + 10 \times ( \text{STDNorm} + \text{STDAngle}).$$
>
> The choice of the hyperparameter here is to make NC1 (CDNV) and NC2 (STDNorm+STDAngle) have a more balanced scale. We use the AdamW optimizer with learning rate 1e-4 and weight decay value 1e-6, and Cosine LR scheduler with linear warmup. This optimizer and scheduler choice is consistent with training in the original diffusion policy paper. We pretrain the vision encoder for 50 epochs and then end-to-end train the entire pipeine (i.e., together with the diffusion model action decoder) for another 50 epochs.

---

> ### Author Response · Authors · 2024-11-17
> **Response to Reviewer NdZb (II)**
>
> ### **[C] "Why does the clustering phenomenon emerge for the planar pushing task with a continuous control signal? The toy example has discrete states in the closed-form optimal policy, so one can see why the clustering phenomenon might arise. The planar pushing task presents a different scenario where this emergence seems less intuitive."**
>
> This is a great question. We don't think we can provide a rigorous explanation at this moment, which will perhaps need theoretical analysis such as those in [paper 1](https://arxiv.org/abs/2110.02796), [paper 2](https://arxiv.org/abs/2105.02375), and [paper 3](https://arxiv.org/abs/2207.10541).
>
> However, we can provide an intuitive explanation:
>
> Consider two types of input images in push-T. In type I, the object T needs to be pushed to the **left** to the target, while in type II, the object T needs to be pushed to the **right** to the target. Consider the ResNet features of both types of inputs -- the features of those two types of images must be **far away** from each other to gain **robustness**.
>
> If the features were very close to each other, then given a new input image, the features of the new image may be similar to both features of type I and type II -- hence it is **ambiguous** for the network to decide whether the object should be pushed to the left or to the right.
>
> In our reasoning above, we only consider two types of "action modes" (push left and push right). In the case of $K$ types of action modes, in order for each mode to be highly **distinguishable** from the others, the features of the $K$ modes must be pushed far away from each other, simultaneously. The Simplex ETF geometric configuration happens to be a configuration of the $K$ modes such that each mode is far away from the rest (imagine the case of a perfect tripod when $K=3$).
>
> So in summary, what clustering entails is that during training time, the network learns to push the features of each "action mode" far away from each other, so that any given new sample would clearly beong to one of the $K$ modes. (Of course $K$ here can be very large for complex tasks, which could make the clustering very difficult.)

---

> ### Author Response · Authors · 2024-11-18
> **Response to Reviewer NdZb (III)**
>
> ### **[D] "There are no ablation studies provided on the relative importance of the two NC metrics in pretraining. "**
>
> Thank you for this suggestion and we added one specific ablation study on the relative importance of the NC metrics (CDNV, STD_Norm, STD_Angle) in pretraining. We show the result of sweeping over different ratios of the three metrics in pretraining loss, using the 100 demo push-T dataset. From this ablation study, we can see that our pretraining method is not very sensitive to the hyperparameters for the coefficients of each loss term, as long as they are in a reasonable range. This makes our method easier to replicate and being applied for different datasets.
>
> | Coefficient of CDNV |  Coefficient of STD_Norm | Coefficient of STD_Angle | Score
> | -------- | -------- | -------- | -------- |
> | N/A (Baseline)  |  N/A (Baseline) | N/A (Baseline) | 24.1% |
> | 0.1 | 10 | 10 | 35.1% |
> | 1| 10 | 10 | 35.2% |
> | 1 |1 |1 |35.7% |
> | 1 |1 |10 | 36.1%|
> | 1 | 1000| 1000| 31.9%|

---

> ### Comment · Reviewer_NdZb · 2024-11-26
> **Thanks for the rebuttal and additional results!**
>
> Thank you for addressing my questions and the additional ablation study! The provided responses helped the readers better understand the approach and its performance. After reading the authors' responses and comments from other reviewers, I still think this is an excellent paper with novel and interesting findings.

---

### Official Review · Reviewer_FVeR · 2024-11-04

**Soundness:** 3
**Presentation:** 2
**Contribution:** 3
**Rating:** 6
**Confidence:** 3

**Summary:**

This paper investigates whether visual features in image-based robotic control systems exhibit neural collapse (NC) - a clustering phenomenon previously observed in image classification tasks where features group into distinct classes. The authors find that on robotic control task of planar pushing, the visual features do indeed cluster into eight "control-oriented" classes based on relative pose between objects and targets. Most importantly, they show this clustering behavior can be leveraged to improve robot performance by up to 35% in low-data scenarios by pretraining the visual encoder to encourage such clustering, demonstrating benefits in both simulated and real-world robotic pushing tasks.

**Strengths:**

1. I like the interpretable way of converting a regression task (in the context of behavior cloning) of action prediction to (x, y, $\theta$), which can be discretized.

2. It is interesting to see that the authors have included experiments on a real robot for the task of planar pushing and especially the boost that pretraining to minimize NC metrics gives compared to the baseline.

**Weaknesses:**

3. The introduction can have more motivation of what NC is and why and where is this phenomenon observed rather than elaborating too much on the toy example. For instance, in the **Our goal** paragraph (L112), the authors raise the question "Does a similar law of clustering, in the spirit of NC, happen when cloning image-based control policies?" However, it is not very clear at this point on why this clustering is important to begin with? What happens if my control policies are working well and aren't clustered in the spirit of NC? Why should the reader care. Specifically it'd be good if authors can add the following:
	- Explicitly state potential benefits of finding NC in control policies (e.g. improved generalization, more interpretable features)
	- Expand on why clustering in the visual representation space might be important for control policies, even if they are already working well

4. In addition to this the related works is pushed to the appendix giving the reader a hard time to contextualize the current work. I'd highly suggest rewriting the related works as well as the motivation/introduction to address #1.

5. Fig 5 is really hard to read. Can't the plots for the 4 methods (ResNet/DinoV2 + LSTM/DM) be combined into a single plot for each of the 3 metrics (CDNV, STD Angle, STD Norm), resulting in 3 plots?

6. Can the authors visualize (x, y $\theta$) for the 4 methods during testing policy rollouts in a 3D space as that was the main driving point for proposing REPO (L260)?

----
**Rationale for voting**: As of now, because of lack of clarity and potential re-writing on the motivation of this work (points #3, 4) and lack of clear experimental results that interpret the pretrained models with minimizing NC metrics (point #6) and organization of the results (point #5) -- I currently believe that this paper needs more work and hence am voting for a reject. However, my final decision would be based on other reviewers' comments and authors' rebuttal.
---
**Post rebuttal update**: Based on the authors rebuttal, I have increased my score to 6 as they have addressed all my major concerns regarding their paper.

**Questions:**

7. Will the results shown for planar pushing generalize for other robotic manipulation tasks such as Lifting Cube or Stacking cubes? There are several simulators such as Robosuite that provide with demonstrations of such task -- can REPO be extended to these non-planar tasks in a trivial manner or is it a future avenue of research?

8. Some results in Sec 3.1 (finetuning experiments), are not new. Observation (i) is stated as "During retraining on the target domain, performance increases in the target domain but drops in the source domain." -- Isn't this true for deep learning networks in general?

9. Why aren't DinoV2 features also used for Table 1 (i.e pretraining DinoV2 to minimize NC metrics) and using that for control policies?

10. What is the Baseline model architecture that is used in Table 1?

---

> ### Author Response · Authors · 2024-11-16
> **Response to Reviewer FVeR (I)**
>
> We thank the reviewer for appreciating our classification strategy and real-world experiments.
>
> In the next, we address the reviewer's concerns.
>
>
> ### **[A] "What is the Baseline model architecture that is used in Table 1?"**
>
> We apologize that we did not provide sufficient details about the baseline model in the original paper.
>
> To provide more details. For both the baseline and our method, the network architecture is exactly the same -- We used ResNet as the vision encoder and diffusion model (DM) as the action decoder. ResNet encodes images as compact latent vectors, and the latent vectors are used to parameterize the score function for action diffusion via a U-Net. The entire pipeline is trained end-to-end with expert demonstrations using action noise L2 loss supervision. We suggest the reviewer to look at the original [diffusion policy paper](https://diffusion-policy.cs.columbia.edu/diffusion_policy_2023.pdf) if more details are needed.
>
>
> Our method is different from the baseline only in terms of training strategy. We first pretrain the same ResNet using NC regularization (i.e., minimize the NC metrics and encourage the visual features to cluster according to control-oriented labels), and then end-to-end train the ResNet+DM pipeline from expert demonstrations.
>
> We will add these clarifications to the paper.
>
>
>
> ---
>
>
> ### **[B] "Fig 5 is really hard to read. Can't the plots for the 4 methods (ResNet/DinoV2 + LSTM/DM) be combined into a single plot for each of the 3 metrics (CDNV, STD Angle, STD Norm), resulting in 3 plots?"**
>
> Thanks for the suggestion. We want to clarify that our choice of drawing four different methods separately is largely motivated by the [original paper on Neural Collapse](https://arxiv.org/pdf/2008.08186) (see e.g., Figs. 2-4 there). Since then, this has almost become a convention in related works.
>
> Another reason to separate these graphs is that since they use different model architectures, the absolute values of the metrics are generally different. Separating them helps the reader visualize the prevalent **relative decrease of the metrics**.
>
>
> However, to accommodate the reviewer's request, we provide plots combining 4 methods into a single plot for each of the 3 metrics. Please find the plot [here](https://drive.google.com/file/d/1PA6OgG89zsVtfcoqOR2KYxbna9C0jkM4/view?usp=drive_link). We are happy to add this plot to our paper.
>
>
>
> ---
>
>
> ### **[C] "Can the authors visualize $(x, y, \theta)$ for the 4 methods during testing policy rollouts in a 3D space as that was the main driving point for proposing REPO (L260)?"**
>
> Thank you for your question. We would like to properly address your concerns but would appreciate some clarification on the specific aspects you are questioning.
>
> We provide two plots in response to the reviewer's request:
>
> - Visualization of the training samples' $(x,y,\theta)$ trajectories (total 500 expert demos): see [here](https://drive.google.com/file/d/113dpFDZ_vgFI-kcc2GR_x_tUT6yPuSCq/view?usp=sharing)
> - Visualization of the test samples' $(x,y,\theta)$ trajectories (total 100 test cases): see [here](https://drive.google.com/file/d/1URZih6sJT13RZd0GE3r7tklKHEf_HUFi/view?usp=sharing).
>
> We can certainly add these plots to our paper, but we wonder what exactly does the reviewer want to ask about these plots?

---

> ### Author Response · Authors · 2024-11-16
> **Response to Reviewer FVeR (II)**
>
> ### **[D] "Will the results shown for planar pushing generalize for other robotic manipulation tasks such as Lifting Cube or Stacking cubes? ..... can REPO be extended to these non-planar tasks in a trivial manner or is it a future avenue of research?"**
>
> YES! Thanks for suggesting the block stacking task.
>
> We performed experiments on the block stacking dataset and found our findings also hold.
>
> Specifically, we trained the diffusion policy based on a 1000-demo expert block stacking dataset "core stack_d0" downloaded from the [MimicGen documentation website](https://mimicgen.github.io/docs/datasets/mimicgen_corl_2023.html). Here is a [sample demo](https://drive.google.com/file/d/1RZTBHNgESDgE041CQGqE2k_3yd8TPZLk/view?usp=sharing) for the reviewer to take a look.
>
> Similarly as push-T, we posit that the key role of visual representation is to convey a goal to the vision encoder. Therefore, we compute the relative 2D position and the relative rotation angle (around $z$-axis on the table) between two blocks, namely $(\Delta x, \Delta y,\Delta \theta)$ and then divide that 3D space into 8 orthants to classify training samples into 8 different classes.
>
> We plot the three NC metrics as training progresses, see [here](https://drive.google.com/file/d/1PNTYnk8hE5jKi_7L1Q8et_rlCCrTQ-tp/view?usp=drive_link). We clearly observe the three metrics keep decreasing during training.
>
> This experiment shows that control-oriented clustering is quite general in vision-based control.
>
> **Limitation and future work**. However, we do want to admit that, as Reviewer NdZb pointed out, the current classification strategy is interesting but faces scalability challenges in complex manipulation tasks involving many objects or deformable objects (because the number of poses that need to be tracked quickly increases, leading to many classes). A future avenue of research is definitely to design more scalable methods to classify the training samples, especially in complex manipulation tasks.
>
> However, as a first paper studying control-oriented clustering in vision-based control, we believe the experiments in our paper are already quite extensive after incorporating the many useful suggestions provided by the reviewers. We hope the reviewer can agree with us on this, and let us know if there is more we can do.
>
>
>
> ---
>
>
> ### **[E] "Some results in Sec 3.1 (finetuning experiments), are not new. Observation (i) is stated as "During retraining on the target domain, performance increases in the target domain but drops in the source domain." -- Isn't this true for deep learning networks in general?"**
>
> The reviewer is absolutely right -- yes this observation is not new at all and it is generally true for deep learning networks.
>
> We did not mean to claim that observation as new or even as a contribution of the paper (we apologize if we sounded like that in the paper). In that paragraph we were simply stating our observations. The main point we wanted to make there was the observation (ii) of "neural re-clustering" and it seems to correlate with observation (i) -- suggesting that the degraded/improved performance on the source/target domain may be related to the re-clustering phenomenon. We hoped that observation could potentially motivate more in-depth study and analysis in the future.
>
>
>
> ---
>
>
> ### **[F] "Why aren't DinoV2 features also used for Table 1 (i.e pretraining DinoV2 to minimize NC metrics) and using that for control policies?"**
>
> The reason why we only performed control-oriented visual pretraining for ResNet is twofold.
>
> - ResNet and DINOv2 deliver quite similar performance on the planar pushing task, as shown in Fig. 4 of our paper (DINOv2 performs slightly better, but not very significant).
>
> - Since DINOv2 is larger than ResNet, it takes significantly more resources and time to experiment with DINOv2. For example, finetuning DINOv2 is almost 10 times slower than finetuning ResNet.
>
> However, to accommodate the reviewer's request and make our results more extensive, we have re-done experiments in Section 4 (and Table 1) using DINOv2, with 100 demonstrations.
>
> The following table shows the results.
>
> | Pipelines | Domain R | Domain O | Domain B | Domain T |
> | -------- | -------- | -------- | -------- | -------- |
> | Baseline (w/ DINOv2)    |  18.95%    |   37.47%   |  24.63%   |      19.42%      |
> | Control-oriented pretraining (w/ DINOv2) | 21.10% | 46.02% | 24.66% | 24.68% |
>
> Clearly, we observe that control-oriented pretraining improves test-time performance, regardless of whether ResNet or DINOv2 are used.

---

> ### Author Response · Authors · 2024-11-16
> **Response to Reviewer FVeR (III)**
>
> ### **[G] "lack of clarity and potential re-writing on the motivation"**
>
> The reviewer raised a major concern about the style and clarity of presentation. Below are some of our thoughts:
>
> - There is no presentation style that is universally appreciated by everyone. For example, Reviewers 93jv and NdZb seem to particularly like our unconventional writing style and find it engaging. We'd like to give the reviewer reasons about why we decided to spend considerable text on the toy problem:
>     - Two major types of audience of our paper are those from neural collapse (representation learning) and those from vision-based control (robotics, imitation learning etc). However, the former community is usually not familiar with the setup of imitation learning in vision-based control, and the latter community is in general not familiar with the concept of neural collapse.
>     - Therefore, to make sure both communities are on the same page, we used the toy problem of double integrator to "synchronize" two communities: the representation learning community will understand neural collapse in the "fresh" new example of control; and the robotics community will "graphically" see the neural collapse phenomenon from the toy example they are very familiar with.
>
> We believe this toy example is quite new and unique for both communities (indeed, it took us a while to carefully design this toy example). Hopefully our explanation helps the reviewer understand our choice.
>
> - In the meanwhile, we **absolutely agree with the reviewer** that our introduction falls short of providing a strong motivation for observing and understanding NC in vision-based control. To address this, we will take the suggestions offered by the reviewer. Particularly, there are two strong motivations for studying NC in vision-based control:
>     - **Fundamental understanding: machine learning interpretability**. Understanding representation learning is crucial for us to improve the interpretability of ML and DL models, even if the current models already work very well. This is indeed one of the key topics in ICLR. For example, [one related work](https://openreview.net/forum?id=w1UbdvWH_R3) on Neural Collapse won the [Outstanding Paper Award at ICLR 2022](https://blog.iclr.cc/2022/04/20/announcing-the-iclr-2022-outstanding-paper-award-recipients/). In the context of vision-based control, **the visual representation space is the information channel from perception to control**, and therefore understanding visual representation (and how it differs from the visual representation in vision tasks) is central in expanding our understanding of vision-based control.
>     - **Practical algorithm: improving generalization**. Unlike image classification that is considered more or less a solved problem in computer vision, **vision-based control is far from being solved** in robotics and related fields. As shown in our experiments, behavior cloning is data hungry -- it delivers poor performance under insufficient demonstrations. Our paper showed that, by encouraging control-oriented clustering in the visual representation space, test-time performance can be significantly improved, using the same amount of demonstrations.
>
> We will add discussion of these motivations to our paper and revise its introduction and related works.
>
> We plan to submit a revision of our paper a few days before the rebuttal period ends, including new experiments and discussions -- we want to wait until we received all feedback from the reviewers so our revision can incorporate as many reviewer suggestions as possible.
>
> Let us know if this is fine to the reviewer!

---

> > ### Comment · Reviewer_FVeR · 2024-11-22
> > **Thanks for the rebuttal and experiments!**
> >
> > I thank the authors for their detailed comments and experiments that they have conducted during the rebuttal phase.
> >
> > [A] Thanks for clarifying the baseline model. Yes, addition of these details would make it clear to the reader that the baseline architecture are indeed a fair comparison.
> >
> > [B] I can see where the authors are coming from and I leave it upto the authors to decide which plot to include. Thanks for combining the plot and yes I did initially notice that the absolute values or the metrics evaluated are indeed different, but the combined plot provides an easy way to compare across models.
> >
> > [C] It does appear that the test rollout policy has a different distribution over $\Delta \theta$ -- is that because of the different intial configuration during the test time?
> >
> > [D] Thanks for adding this experience on the challenging task of Stack Cube. I do have a follow up question:
> >
> > The model works well for action spaces that are typically well defined in the 3D cartesian space -- for robotic manipulation that would be the End-effector control. I am wondering if one changes the action space to a 8-dimensional delta joint angles -- would the proposed method still work well?
> >
> > [E] Thanks for the clarification.
> >
> > [F] Thanks for adding the DinoV2 results. You can consider adding it either in the main paper or in the appendix for the sake of completeness.
> >
> > [G] Thank you for the detailed answer. I do think adding the two strong motivations would be **incredibly helpful** to readers who are not super familiar with the NC background and it will make a stronger case for the paper.
> >
> > Yes, sending the revised draft of the paper at a closer date of the rebuttal's end would be fine by me. Thanks for asking.
> >
> > ----
> >
> > I do have an additional question to the authors' responser to the concern raised by Reviewer **Nat5**. Specifically, __Concern [A] "It seems unclear if the downstream improvement is from the NC regularization itself, or from access to ground truth state (by proxy of the class labels)."__
> >
> > In the **Discussion** section, the authors say that the reason why adding GT pose or REPO lables did not help is because of the differing dimensions of the ResNet features (512-dim) and the GT (20-dim). For a fair comparison, shouldn't the GT pose be encoded to a slighly higher dimension (using a 2-3 layer MLP) such as 128-dim or 256-dim (need to perform some hyperparameter tuning here)?
> >
> > ----
> > **Revised score**
> >
> > I do think the authors have provided satisfactory responses to my concerns and hence I am increasing my score to accept with a score of 6.

---

> > > ### Author Response · Authors · 2024-11-26
> > > **Response to Reviewer FVeR (IV)**
> > >
> > > We thank the reviewer so much for appreciating the value of our response.
> > >
> > > ### **[C] "It does appear that the test rollout policy has a different distribution over $\Delta \theta$ -- is that because of the different intial configuration during the test time?"**
> > >
> > > Thank you for your clarification. Here we want to explain our test-time experiment setup into the following two points:
> > >
> > > - The initialization of $(x, y, \theta)$ for the pushed block and the target pose uses the same randomization function during both training and test time. The only difference is that we use different random seeds to initialize training and test demos to make sure the trained policy is tested under unseen cases.
> > > - We completely agree with your observation that the distribution over $\Delta \theta$ for test rollout samples is different to the distribution for training samples. But this is caused by the trajectories predicted from the trained baseline diffusion policy. The large variance of $\Delta \theta$ indicates that the baseline policy is not as perfect as expert training demos.
> > >
> > > ### **[D] "Would the proposed method still work well if one changes the action space to a 8-dimensional delta joint angles?"**
> > >
> > > Thank you for this thoughtful question. We would like to clarify that the Franka Panda arm, which is used in the block stacking task, has 7 degrees of freedom (7 joints) by design.
> > >
> > > Instead of labeling the training samples based on the objects' relative pose in the 3D cartesian space, we provide a 128-class labeling metric based on the signs of the 7-dimensional delta joint angles. We plot the three NC metrics using this 128-class labeling [here](https://drive.google.com/file/d/1VHiP-a8vlpxrn9H0fVrd3EMe74AdJ7sM/view?usp=drive_link), where both "CDNV" and "STD Norm" curves indicate that using 7-dimensional delta joint angles does not show the control-oriented clustering phenomenon.
> > >
> > >
> > >
> > >
> > > ### **[H (Additional question)] "For a fair comparison, shouldn't the GT pose be encoded to a slighly higher dimension (using a 2-3 layer MLP) such as 128-dim or 256-dim (need to perform some hyperparameter tuning here)?"**
> > >
> > > This is a great question.
> > >
> > > We provide a supplemental experiment for the new results in the bottom four lines of the table below. We observe that:
> > >
> > > - Yes, encoding the GT block and target pose into a dimension similar to the image embedding **increases the performance** compared to baseline, and baseline with ResNet representation directly concatenated with GT poses.
> > > - However, the performance is **still not as good as our NC regularization method**.
> > >
> > >
> > > Moreover, we want to emphasize that this is not a fair comparison to our method, also mentioned in the response to Reviewer Nat5.
> > >
> > > This is because baseline and our control-oriented pretraining method do **NOT** use any information about groundtruth states at **TEST** time. **At test time, the pipeline is still a pure vision-based control pipeline.** This is not the case for all the other pipelines suggested by the reviewers (in the last six rows of the Table), which all require access to GT states.
> > >
> > > | Pipelines | Performance on push-T |
> > > | -------- | --------  |
> > > | Baseline     | 24.1%     |
> > > | Control-oriented pretraining (goal-based) | 34.7% |
> > > | Control-oriented pretraining (action-based) | 35.1% |
> > > | ResNet representation + REPO label | 20.0% |
> > > | ResNet representation + GT block and target pose| 22.3% |
> > > | **ResNet representation + 256 dim GT block and target pose (via 3 layer MLP)** |26.2%  |
> > > | **ResNet representation + 128 dim GT block and target pose (via 3 layer MLP)** |21.8%  |
> > > | **ResNet representation + 256 dim GT block and target pose (via 2 layer MLP)** |26.7%  |
> > > | **ResNet representation + 128 dim GT block and target pose (via 2 layer MLP)** | 26.8% |

---

### Official Review · Reviewer_Nat5 · 2024-11-04

**Soundness:** 3
**Presentation:** 3
**Contribution:** 4
**Rating:** 8
**Confidence:** 4

**Summary:**

This paper investigates whether the phenomenon of neural collapse, previously observed in image classification and other tasks, appears in visual representations trained for continuous control. The paper considers the setup of training a visual encoder end-to-end jointly with a behavioral cloning policy on a planar pushing task. To study it as a classification problem, the frames are binned by relative poses to future states. The paper shows that during end-to-end training, the visual representation undergoes neural collapse. Finally, the neural collapse metrics are used as a regularization objective to improve downstream behavioral cloning performance.

**Strengths:**

- The study of neural collapse in the visuomotor control setting is novel and well-motivated.
- The minimum-time double integrator example is very clear.
- Sim and real experiments show that neural collapse regularization objectives improve downstream task performance.
- Fine-tuning experiments show that neural collapse happens when transferring across domains.

**Weaknesses:**

- It seems unclear if the downstream improvement is from the NC regularization itself, or from access to ground truth state (by proxy of the class labels).
- The criteria for determining whether neural collapse has occurred seems unclear in this setting.
- Experiments are mostly on the planar pushing task.

**Questions:**

- In Table 1, is the performance improvement from access to privileged information (ground truth state), or from the NC regularization itself? It would be good to see a baseline where the orthant class label is provided as input to the policy alongside the image embedding, but no NC regularization is used.
- Could you clarify what constitutes neural collapse when we look at CDNV, STD Angle, STD Norm? In line 293-304, it is stated that these metrics should tend to zero as an indicator of neural collapse. However, in the figures below, none of the metrics seem to tend to zero.
- How sensitive is the observation of neural collapse to the choice of clustering? The hypothesis that the visual representation conveys a goal to the action decoder seems arbitrary. For example, one could also posit that the visual representation encodes the current environment state (block and EE pose). What happens if we cluster on environment state? Do we still observe neural collapse? What if we assign class labels uniformly at random? It would be convincing to see these comparisons as a justification for this choice.
- If the core claim is that neural collapse happens for visual encoders trained end-to-end in control tasks, it could perhaps be clearer and more convincing to apply this first to discrete control, and show that neural collapse in visual representation occurs across various discrete control environments like CartPole, etc.

---

> ### Author Response · Authors · 2024-11-15
> **Response to Reviewer Nat5 (I)**
>
> We thank the reviewer for appreciating the novelty, presentation, and experiments of our paper.
>
> In the next, we address various concerns and questions raised by the reviewer.
>
>
> ### **[A] "It seems unclear if the downstream improvement is from the NC regularization itself, or from access to ground truth state (by proxy of the class labels)."**
>
> This is a great question! To answer it, we performed experiments concatenating either the REPO label or the groundtruth states (i.e., block, target, and end-effector poses) with the ResNet visual representation.
>
> The following table shows performance of several methods on the push-T task (using 100 demonstrations as described in Section 4 of the paper).
>
>
> | Pipelines | Performance on push-T |
> | -------- | --------  |
> | Baseline     | 24.1%     |
> | Control-oriented pretraining (goal-based) | 34.7% |
> | Control-oriented pretraining (action-based) | 35.1% |
> | **ResNet representation + REPO label** | 20.0% |
> | **ResNet representation + GT block and target pose** | 22.3% |
>
> Results in the first three rows are already reported in Table 1 of the original paper, while results in the last two rows are obtained from concatenating REPO labels or GT states to the visual representation. Clearly, adding groundtruth states to the visual representation does not improve the performance.
>
> **Discussion**. This observation is not surprising. Indeed, the ResNet representation is of dimension 512, but the new information concatenated has dimension below 20. Likely, the newly added dimensions will destroy the original structure in the ResNet representation (and typically require some "alignment" between representation spaces). This is like adding apples to oranges.
>
>
> **Remark**. We should emphasize that, the pipeline, even with control-oriented pretraining, **does NOT use any information about groundtruth states at TEST time**. At test time, the pipeline is still a pure vision-based control pipeline. This is not the case for the pipelines suggested by the reviewer (in the last two rows of the Table), which both require access to GT states. This is a crucial clarification.
>
> ---
>
> ### **[B] "The criteria for determining whether neural collapse has occurred seems unclear in this setting.""**
>
> The reviewer raised two questions in this concern, and we address them separately below.
>
> - **"Could you clarify what constitutes neural collapse when we look at CDNV, STD Angle, STD "Norm"?**
>
> CDNV characterizes the ratio between within-class variation and between-class variation. Intuitively, imagine features of each class form a point cloud, CDNV evaluates how far are these point clouds (belonging to different classes) get pushed away from each other.
>
> STD Angle measures the standard deviation of the set of pairwise angles spanned by globally centered class means. For example, in Fig. 1 of our paper, it is the standard deviation of the three angles in the red band (evaluated at every epoch).
>
> STD Norm, similarly, measures the standard deviation of the set of lengths of globally centered class means. For example, in Fig. 1, it is the standard deviation of the three lengths in the blue band (evaluated at every epoch).
>
> (NC1) is equivalent to CDNV equals to 0, and (NC2) is equivalent to STD Norm and STD Angle both equal to 0.
>
> - **"it is stated that these metrics should tend to zero as an indicator of neural collapse. However, in the figures below, none of the metrics seem to tend to zero."**
>
> Yes the reviewer is correct. Neural Collapse (NC), or perhaps better stated as "Perfect Neural Collapse", happens when all three metrics become zero.
>
> In our experiments, all the metrics decrease as training progresses, but they never become exactly zero. This is the reason why in our paper we keep mention "clustering" (even in our paper title) to state that our observation and analysis of the clustering is inspired by pioneering work on NC, but our phenomenon is **only "clustering" but not "collapsing"**. The visual features tend to cluster, but we do not observe perfect NC.
>
> Nevertheless, even though there is no perfect NC in the visual representation space, pretraining the visual features by minimizing NC metrics (and hence encourage NC) does lead to significant performance improvement, as shown by our experiments.
>
> We apologize if this point has been confusing to the reviewer. We did not want to invent our own name of the phenomenon and want to pay enough respect to prior work. We will further emphasize this point in our paper.

---

> ### Author Response · Authors · 2024-11-15
> **Response to Reviewer Nat5 (II)**
>
> ### **[C] "How sensitive is the observation of neural collapse to the choice of clustering?"**
>
> Thanks for the question and suggesting other classification strategies.
>
> We should note that in our original paper we already provided a different classification strategy based on $k$ means clustering and showed that it did not lead to observation of clustering and neural collapse (at least much weaker than the clustering strategies designed in the paper).
>
> We tried the two classification strategies suggested by the reviewer. (a) The block has three states $(x,y,\theta)$ and the end effector (EE) has two states $(x_e,y_e)$. In total we have five states. We divide every state dimension into two bins, leading to $2^5=32$ classes. (b) We assign 8 classes uniformly at random to each training sample.
>
> The following table summarizes the results (please click the hyperlink to see the plots).
>
> | Classification Strategy | Plots of NC Metrics  |
> | -------- | -------- |
> | 32 classes based on environment state (block + EE)     | [plot](https://drive.google.com/file/d/1z9oi9gdfabg9i9deCbPH0x_iQx6k7GLz/view?usp=drive_link)     |
> | 8 classes uniformly at random | [plot](https://drive.google.com/file/d/1rqkNdRQQLiF4iAesul7fySFSWdOqVcgg/view?usp=drive_link) |
> | 5 classes based on $k$ means clustering | see Appendix A Fig. 13 |
>
> Clearly, the clustering or NC phenomenon is much weaker compared to what we have shown in Fig. 5 of our paper.
>
> Please let us know if the reviewer wants to try other classification strategies.
>
>
>
> ---
>
>
> ### **[D] "it could perhaps be clearer and more convincing to apply this first to discrete control, and show that neural collapse in visual representation occurs across various discrete control environments like CartPole, etc"**
>
> This is a great suggestion. We are currently performing extra experiments on discrete-control tasks and will report our findings in a separate thread.
>
> In addition, to make our experiments more extensive, and as suggested by Reviewer FVeR, we have conducted an **extra experiment on the block stacking dataset**, please refer to **Part [D] in our response to Reviewer FVeR**!

---

> ### Author Response · Authors · 2024-11-17
> **Response to Reviewer Nat5 (III): Vision-based Discrete Control -- Lunar Lander**
>
> To accommdate the reviewer's request and perform an experiment where the control is discrete, we choose a vision-based lunar lander problem.
>
> **Lunar lander**. The lunar lander task is an [environment in OpenAI Gym](https://gymnasium.farama.org/environments/box2d/lunar_lander/), whose control space is discrete with 4 actions:
> - 0: do nothing
> - 1: fire left orientation engine
> - 2: fire main engine
> - 3: fire right orientation engine
>
> **Expert demonstration collection**. To collect an expert demonstration dataset on the lunar lander task, we trained an expert reinforcement learning algorithm -Proximal Policy Optimization (PPO)- on the [LunarLander-v2](https://www.gymlibrary.dev/environments/box2d/lunar_lander/) gym environment. We applied the default optimal network architecture and training hyperparameters from the popular RL training library -- [rl_zoo3](https://github.com/DLR-RM/rl-baselines3-zoo) to train the PPO. Then we used the trained PPO policy to collect 500 demos as our expert training dataset. The reviewer can check a sample collected demo from [here](https://drive.google.com/file/d/1ppU2mtgj-bjFGPgmdXIpE3jS2DjRjGsS/view?usp=sharing).
>
>
>
> **Behavior cloning**. After collecting expert demonstrations, we train an imitation learning model in this way: We use a ResNet 18 model as the vision encoder and use an MLP model to map the 512 dimensional latent embedding into 64 dimensional. We use a sequence of 4 input images as observation input and predict the next action. The action decoder is structured as another MLP network with 6 layers and maps the image latent embeddings to predicted action, then calculating the cross-entropy loss with the ground truth discrete actions.
>
>
> **Control-oriented clustering**. We plot the three NC metrics with respect to training epoches in [this figure](https://drive.google.com/file/d/19Kfo1YII-Ubq97AsgQiCER4im1MxQnOX/view?usp=drive_link). **We observe very strong control-oriented clustering in the visual representation space**.
>
> **This experiment further reinforced the universality of control-oriented clustering in visual latent representation.**
>
> **Why not Cart-pole?** It is worth explaining why we did not choose the [cart-pole environment in OpenAI Gym](https://www.gymlibrary.dev/environments/classic_control/cart_pole/). *This is because the cart-pole environment only has two discrete actions, which is trivial for neural collapse.* To see this, observe that equation (4) of our paper, in the case of two classes (two discrete actions), becomes:
>
> $$
> \mu = \frac{1}{2}(\mu^1 + \mu^2),
> $$
>
> which is equivalent to
>
> $$
> \frac{1}{2}(\mu^1 - \mu) + \frac{1}{2}(\mu^2 - \mu) = 0
> $$
>
> where $\tilde{\mu}^1:=\mu^1 - \mu$ and $\tilde{\mu}^2:=\mu^2 - \mu$ are the globally centered class mean vectors. From the above equation we see that $\tilde{\mu}^1$ and $\tilde{\mu}^2$ must be **antipodal** -- a trivial clustering in the case of two classes (regardless of what the neural network weights are).
>
> Therefore, the minimum number of classes we need for a nontrivial (and non-vacuous) observation of neural collapse is three -- hence we chose Lunar Lander with 4 discrete actions.

---

> > ### Comment · Reviewer_Nat5 · 2024-11-18
> > **Thank you for the additional experiments and insights**
> >
> > Thank you for the additional experiments and insights! My concerns have been addressed, and overall I am positive about this paper. It is clear from the LunarLander experiment that neural collapse occurs for discrete visuomotor control, and the paper shows that this phenomenon also occurs for continuous control, up to some choice of discretization, with nonlinear policy heads such as diffusion. I think is an interesting result by itself. The paper then shows that NC regularization helps with downstream policy performance. This opens up further research as to what kind of action discretization corresponds to a strong neural collapse in a continuous setting, whether collapse happens in the intermediate representations of the policy head, and whether NC regularization can be applied to a broader setting without ground truth labels, etc.

---

### Official Review · Reviewer_93jv · 2024-11-04

**Soundness:** 3
**Presentation:** 3
**Contribution:** 3
**Rating:** 8
**Confidence:** 3

**Summary:**

This paper studies neural collapse for control tasks. To facilitate this study, the authors learn policies with behavior cloning end-to-end. To probe the learned representations, they define a heuristic classification of features learned within the encoder called REPOs that categorize the position and orientation of the object in the image relative to the target object (this is specific to planar push tasks). They find that pretraining the vision encoder using neural collapse as a regularization technique improves performance on a simulated and real-world push-t task. They also find that, when the model is continually fine-tuned on different planar push tasks, the visual representation space undergoes re-clustering instead of saturating on early training tasks.

**Strengths:**

To my knowledge, this is the first paper to study neural collapse in control. I appreciated some of the unconventional writing choices. The introduction had a nice unification of optimal control and behavior cloning and made the relevance of neural collapse to control very apparent. The experimental approach was really unique, creative, and insightful. The continual learning findings were especially interesting from my perspective.

**Weaknesses:**

The paper could benefit from a more concise abstract and a more broad introduction that provides stronger motivation for the problem and clearer articulation of the paper's contributions. I like the unconventional approach to the abstract, but it would be more clear to the audience if the toy experiment appeared as a preliminary to the method section.

I took issue with the discussion of resnet features throughout the paper.
- First in response to L213: pre-trained resents are still useful compared to randomly initialized models, and second, that fine-tuning is often required for many transfer learning tasks, so I don't find it surprising that we need to fine-tune imagenet trained resnets for control domains.
- Because the image input of push-t is so limited it's unfair to make broad claims about the utility of resnet features. The benefit of resnet features will be more apparent in more visually challenging settings (e.g., cluttered table-top manipulation).

Overall, as a study of visual representations the findings need to be verified on more visually complex domains.


Typo: "Expect" instead of "expert" on line 118.

**Questions:**

Can you validate these findings on more visually complex domains?

---

> ### Author Response · Authors · 2024-11-16
> **Response to Reviewer 93jv (I)**
>
> We thank the reviewer for appreciating the novelty of our paper in studying neural collapse in vision-based control.
>
> The reviewer had a major concern in ResNet features and visual complexities. We have conducted extensive experiments to address this, detailed below.
>
> ---
>
> ### **[A] "pre-trained Resnets are still useful compared to randomly initialized models"**
>
>  We believe the reviewer's insight about the advantage of pretrained ResNet may be correct for vision tasks. However, whether or not it holds for control tasks needs investigation (we are not aware of such a systematic investigation in the literature, please share evidence/references if possible).
>
>  To investigate this, we compare the performance of four pipelines on the push-T task with a **clean background** (shown in Fig. 2 of our original paper). The following Table shows the results.
>
> | Pipelines | Performance on push-T with clean background|
> | -------- | -------- |
> | **Frozen** *Pretrained* ResNet + DM (Diffusion Model)    |   30.17%   |
> | **Frozen** *Random* ResNet + DM         |   29.95%   |
> | **Finetuned** *Pretrained* ResNet + DM  |   62.83%   |
> | **Finetuned** *Random* ResNet + DM      |   63.66%   |
>
>
> In the table, "Frozen" indicates the ResNet weights are fixed and not finetuned in behavior cloning, while "Finetuned" indicates the weights are end-to-end finetuned from expert demonstrations.
>
>  "Pretrained" indicates the ResNet weights are initialized from pretraining on ImageNet, and "Random" indicates a random set of initial weights.
>
>  In all cases, the action decoder, i.e., the diffusion model (DM) is not frozen and trained end-to-end.
>
>  **Observation in push-T with clean background**. Clearly, the ImageNet-pretrained ResNet does not show benefits either when frozen or finetuned, compared to a set of random weights. [This plot](https://drive.google.com/file/d/1FGYQNP0ostIlK2rsx20pAR-Ayvu8SEwI/view?usp=sharing) shows the training loss curves of finetuning a pretrained ResNet compared with finetuning a randomly initialized ResNet in push-T. They look almost identical (finetuning from a pretrained ResNet is not faster than finetuning from a random initialization).
>
>  With this observation in mind, we then conducted another experiment with a visually complex background, discussed below.
>
> ---
>
> ### **[B] "Because the image input of push-t is so limited it's unfair to make broad claims about the utility of resnet features. The benefit of resnet features will be more apparent in more visually challenging settings (e.g., cluttered table-top manipulation)."**
>
> To study whether our observations holds true for visually complex scenes. We conduct the following experiment.
>
> We replace the clean background of push-T with a **messy table-top manipulation background** adapted from the well-known [LineMod dataset](https://paperswithcode.com/dataset/linemod-1). An example of the generated expert demonstrations can be found [here](https://drive.google.com/file/d/1TkoOkjbpRc2N3R1xe0hSIWi57uwWgIfd/view?usp=sharing). This is not a super realistic scene, but it is a good proxy for visual complexities and is faster for us to test. The following Table shows the results. The pipelines are the same as those in the Table of Part [A].
>
> | Pipelines | Performance on push-T with LineMod background|
> | -------- | -------- |
> | **Frozen** *Pretrained* ResNet + DM (Diffusion Model)     | 32.90%     |
> | **Frozen** *Random* ResNet + DM         |    32.22%  |
> | **Finetuned** *Pretrained* ResNet + DM  |   56.10%   |
> | **Finetuned** *Random* ResNet + DM      |   46.67%   |
>
> **Observation in push-T with complex background**. We make two observations:
>
> - When the ResNet is frozen, regardless of whether pretrained from ImageNet or randomly initialized, the performance is very low. This shows that, if finetuning is not allowed, pretraining on ImageNet does not help.
>
> - However, **when finetuning is allowed**, **pretraining on ImageNet significantly increases the performance**. This shows the rich priors learned from ImageNet indeed helps transfer learning for control tasks, when the scenes are visually complex.
>
>
> **Conclusion from Parts [A] and [B]**. Combining the results from Part A and Part B, we draw the following conclusions to answer the reviewer's question.
>
> - In control tasks, if ResNet is not allowed to be finetuned, then neither pretraining on ImageNet nor random initialization is sufficient to learn a good policy for the control task.
>
> - When ResNet is allowed to be finetuned:
>     - If the scene is visually simple (such as a clean push-T), then pretraining on ImageNet does not show clear advantages over a random initialization.
>     - If the scene is visually complex (with a complex background), then pretraining on ImageNet significantly benefits transfer learning.
>
> We will add the experiments above to the paper. Hopefully the new experiments have answered the reviewer's concern. Please let us know if there is more we can do.

---

> ### Author Response · Authors · 2024-11-16
> **Response to Reviewer 93jv (II)**
>
> ### **[C] "I don't find it surprising that we need to fine-tune imagenet trained resnets for control domains."**
>
> We agree with the reviewer that it is popular practice to finetune ResNet features for downstream tasks.
>
> Perhaps we used too strong words such as "mysterious and counterintuitive" in the paper which irritated the reviewer. We will rephrase the sentences there.
>
> However, what we really wanted to convey there (and a big motivation for our study) is:
>
> - It is generally believed that pretrained ResNet features are "rich" in that they contain information such as corner/edge detection. Under this belief, it is expected that, for visually simple scenes such as pust-T, the ResNet features should contain information about the boundary of the T block and the target, and hence also their position/orientation. This information should be sufficient to push the T-block to the target position (imagine humans do this). However, this is not true from our experiments. Indeed, if one freezes the ResNet and does not allow it to be finetuned with the action decoder, then the performance of the pipeline is very low, as shown in the previous tables (even in visually complex scenes).
>
> - If the ResNet is allowed to be finetuned, then the pipeline works very well. This is very intriguing when viewed from the following perspective: the pretrained ResNet maps input images in the training dataset to a point cloud in the latent space (call it F1), so does the finetuned ResNet (call the point cloud F2). The action decoder is not able to decode the actions from F1 but able to do so from F2. From a distribution matching perspective (since the diffusion model predicts conditional distributions), it suggests that F1 and F2 are different, and the finetuning process, for some reason, changed the mapping from F1 to F2.
>
> So **the fundamental pursuit of our paper is to study in what ways has F2 changed compared to F1?** Our results show that F1 (from ResNet pretrained in ImageNet) is clustered according to **"ImageNet Classes"** (such as dogs and cats, which is well-known from the literature of neural collapse in image classification), but F2 (from ResNet finetuned by push-T) is clustered according to **"control-oriented classes"** (such as push southwest, rotate clockwise).
>
> Of course our finding does not fully characterize the changes from F1 and F2, but we believe it is a step towards understanding the representation learning in vision and control.
>
> Does this make sense to the reviewer? We are happy to elaborate more!
>
>
>
> ---
>
> ### **[D] "Can you validate these findings on more visually complex domains?"**
>
> To validate our findings on more visually complex domains, we perform simulated push-T with visually complex LineMod background (see Part [B] for description and link to demo),
>
> We used 500 demonstrations to train an end-to-end pipeline using ResNet as the vision encoder (pretrained from ImageNet) and Diffusion Model as the action decode. As shown in the second table above, the test-time performance is 56.1%.
>
>
> **Control-oriented clustering under visually complex scences**. We plot the neural collapse (NC) metrics with respect to epoches in [this figure](https://drive.google.com/file/d/1JO0uSAoU4LIpSb6f_A6BEQJV-QLeEOz4/view?usp=drive_link). Clearly, we see the NC metrics decrease as training progresses, suggesting the emergence of clustering in the visual latent space.
>
> This shows that control-oriented clustering still holds even when the scenes are visually complex.
>
> Please let us know if you have questions about the experiments.

---

> > ### Comment · Reviewer_93jv · 2024-11-18
> >
> > I greatly appreciate that you took the time to run these additional experiments and I think the addition of new experiments is a solid improvement on the existing experimental set.
> >
> >  I still think the benchmarks don't reach the necessary minimum for a conference paper. For example, The (Un)Surprising Effectiveness of Pre-Trained Vision Models for Control (https://arxiv.org/pdf/2203.03580) considers similar questions across 4 control domains spanning locomotion and table top manipulation.
> >
> > The ResNet features begin to be performant over random in the fine-tuned setting in [B], which indicates to me that the previous experimental set is not the best proxy for more realistic control tasks. I think this is a really cool paper and I would love to see the results published, but at a future venue with a broader experiment set.

---

> > > ### Author Response · Authors · 2024-11-18
> > > **Response to Reviewer 93jv: broader experiment set**
> > >
> > > Dear reviewer 93jv,
> > >
> > > Thanks for appreciating the value of our response.
> > >
> > > Perhaps it is not clear from our response, but we would like to draw your attention that we actually expanded our experiment set (by a lot) and provided results in response to other reviewers.
> > >
> > > To summarize, now our expanded experiment set contains:
> > >
> > > - **Simulated planar pushing**, both with clean background and visually complex background from LineMod dataset
> > >
> > > - **Block stacking from MimicGen**, see results in Part [D] in our response to Reviewer FVeR
> > >
> > > - **Lunar lander from OpenAI Gym**, see results in response to Reviewer Nat5 (III)
> > >
> > > - **Real-world push-T**, already in our paper.
> > >
> > > Among those, we have three simulated experiments, and one real-world experiment.
> > >
> > > In addition to these four, we are conducting another real-world experiment:
> > >
> > > - **Real-world push-T with visually complex scenes**, where we place many random objects in the table, you can find a sample video [here](https://drive.google.com/file/d/1YgzVMIObA9E2VUoxDN_0eznKcXF_raGJ/view?usp=sharing).
> > >
> > > Therefore, when the new real-world experiment is finished, we would have **five experiments** (three simulated, and two real-world). Note that the paper you kindly shared only contains simulated experiments.
> > >
> > > Lastly but not the least, we would like to emphasize that studying the power of pretrained features such as ResNet is not even the main focus/contribution of our paper -- our paper focuses on the control-oriented clustering of visual representation in various control tasks, which is not in conflict with the paper you shared.
> > >
> > > We don't think it is fair to criticize our paper as "lack of experiments". Please let us know if you have further questions.

---

> ### Comment · Reviewer_93jv · 2024-11-25
>
> Thank you to the authors for engaging in discussion. I appreciate the time that put in to address my concerns and feel they were sufficiently addressed, so I raised my score. My primary concern was the discussion and framing of pre-training. I agree with the authors that this is not the main point of this submission and I think the new framing and expanded experiments set makes the submission quite strong.

---

### Author Response · Authors · 2024-11-26
**Summary of Formal Revision**

Dear Reviewers,

Please allow us to express our sincere gratitude to you for providing extensive and constructive feedback, as well as increasing your ratings. It has been a productive and insightful rebuttal period.

We have incorporated your suggestions and the extra experiments we performed during the rebuttal period into our paper, and here we submit a formal revision.

All the changes we made in comparison to our original paper have been marked in **blue**.

We want to summarize several major changes we made:

- **A more extensive set of experiments** (thanks to the reviewers' great suggestions!)
    - As shown in Fig. 2, we have revised our overview figure to include the discrete Lunar Lander experiment, and the continuous block stacking experiment (in addition to our original planar pushing experiments). In all these simulated experiments, we observe strong control-oriented clustering.
    - In Section 5, we have added an extra set of **real-world planar pushing experiments with visually complex background** (by placing random objects in the scene). As expected, our control-oriented pretraining still leads to substantial improvement over the baseline.


- **Clarity**
    - We have shortened our abstract to be more concise.
    - In the introduction section, "Our goal" paragraph, we have added contents to better inform the reader about the motivation to investigate visual latent representation.

- **Limitations**: in the conclusion section, we have added a paragraph "limitations and future work" to state the limitations of our paper.

- **Appendics**: we have moved many of the interesting rebuttal experiments, and some of the contents in the original main text (which are now lower priority) to the appendics.

In addition to these, we have made minor changes to the structure of the paper to improve the overall flow of presentation.

Please let us know if you have further suggestions about improving the quality of our paper.

Thanks,

Authors

---

### Meta-Review · Area_Chair_yj5e · 2024-12-18

**Metareview:**

The reviewers are generally enthusiastic about this work, highlighting several strengths and providing positive ratings. Notable strengths include being the first paper to study neural collapse in control, employing a unique, creative, and insightful experimental approach, and conducting real robot experiments. Additionally, the presentation of this paper is clear and engaging. The paper received 8,8,8,6 ratings. The AC checked the paper, the reviews and the author responses and the discussions. The AC is in agreement with the reviewers and recommends acceptance.

**Additional Comments On Reviewer Discussion:**

The reviewers engaged in a detailed discussion with the authors, during which the authors clarified several points in the rebuttal and discussion period. These clarifications included an expanded set of experiments, such as simulated planar pushing with both clean and visually complex backgrounds, explanations on changing the action space, and improvements to the clarity of the motivation. Overall, the reviewers found the rebuttal convincing and remained enthusiastic about the work. The AC believes that incorporating more complex tasks is necessary to make the conclusions more generalizable.

---

### Decision · Program_Chairs · 2025-01-22

Accept (Spotlight)